# Double Machine Learning for Causal Inference under Shared-State Interference

**Chris Hays** [1]  **Manish Raghavan** [1]

## Abstract

Researchers and practitioners often wish to measure treatment effects in settings where units interact via markets and recommendation systems. In these settings, units are affected by certain *shared states*, like prices, algorithmic recommendations or social signals. We formalize this structure, calling it shared-state interference, and argue that our formulation captures many relevant applied settings. Our key modeling assumption is that individuals' potential outcomes are independent conditional on the shared state. We then prove an extension of a double machine learning (DML) theorem providing conditions for achieving efficient inference under shared-state interference. We also instantiate our general theorem in several models of interest where it is possible to efficiently estimate the average direct effect (ADE) or global average treatment effect (GATE).

## 1. Introduction

In causal inference, interference — where individuals' treatment assignments, outcomes or other characteristics impact others' outcomes — is everywhere: On short-form video platforms, the consumption of one individual is used as an input to recommendation algorithms that are subsequently used to serve content to others. In ride-sharing or housing rental services, providing a discount to some people can cause them to increase their usage of the service and increase wait times or prices. In such settings, failure to account for interference may lead to biased estimates of causal effects, even in randomized controlled trials.

Across many settings, like those of recommender systems and markets, interference between units often follow a common pattern: the outcomes of individuals depend on others through some *shared state*. In recommender systems, the shared state might be the outputs of recommender systems that are used to generate users' feeds. In marketplaces, the

shared state may be prices, wait times, or availability of inventory. In each of these settings, individuals both influence and are influenced by the shared state.

Assumptions are required to perform causal inference in the presence of interference. Prior work has studied this kind of interference in particular applications (Johari et al., 2021; Basse et al., 2016; Wager, 2024; Munro, 2024; Simchi-Levi and Wang, 2023; Dhaouadi et al., 2023; Farias et al., 2023, 2022; Li et al., 2024). In some cases, they propose either experimental designs or analysis methods that account for interference and allow for valid inference. However, these approaches typically assume a parametric model of the system or domain-specific assumptions.

**Our contributions.** We provide a general framework to reason about causal inference under shared-state interference. We define a formal model in Section 2, where individuals arrive sequentially. Their covariates (i.e., unit characteristics), treatment assignments, and outcomes may influence the outcomes of future individuals through a shared state.

In Section 3, we provide conditions under which efficient inference is possible in the framework. We extend methods from double machine learning (DML) (Chernozhukov et al., 2018), which has shown that it is possible to achieve efficient inference using expressive machine learning methods without parametric functional form assumptions. A key assumption we rely on is that the shared state progresses according to a Markov chain. We also provide a consistent variance estimator, which is necessary for constructing valid confidence intervals and running hypothesis tests.

Next, we instantiate our framework with two applications: first, we show how our framework can be used to estimate average direct effects (ADE) in observational settings; then, we provide a variance reduction strategy for estimation of the global average treatment effects (GATE) in switchback experiments. Intuitively, the ADE measures the expected difference between treatment and control for a unit drawn uniformly at random, keeping the treatment assignment distributions of other units fixed. The GATE measures the mean difference between outcomes when all units are assigned to treatment versus when all are assigned to control. In each instantiation, we provide simulations validating that our method produces estimators that concentrate around

[1]MIT. Correspondence to: Chris Hays <jhays@mit.edu>.

the true treatment effects in finite samples. We also show our consistent variance estimators can be used to construct confidence intervals with the desired coverage probability. We refer the reader to the full version of the paper (Hays and Raghavan, 2025), which we recommend.

### 1.1. Related work.

**Shared-state interference.** Several prior works have explored causal inference in settings captured by or similar to our formulation of shared-state interference (Johari et al., 2021; Basse et al., 2016; Wager, 2024; Munro, 2024; Holtz et al., 2024; Simchi-Levi and Wang, 2023; Dhaouadi et al., 2023; Farias et al., 2023, 2022; Li et al., 2024; Brennan et al., 2024; Bajari et al., 2023; Bright et al., 2023). Some of these papers have analyzed bias resulting from naive estimators or experimental designs and proposed less biased alternatives (Johari et al., 2021; Farias et al., 2022; Brennan et al., 2024). Other works have proposed parametric models of settings that allow for inference in the presence of interference (Wager and Xu, 2021; Li et al., 2024). Several works consider a Markov model of interference similar to ours in an experimental setting and explore less biased estimation methods (Farias et al., 2023, 2022; Glynn et al., 2022).

**Double machine learning.** The double machine learning framework was introduced in Chernozhukov et al. (2018), drawing on a rich literature in semiparametric statistics (see, e.g., Kennedy (2023) for an overview). DML methods have been extended to several other settings, such as those with continuous treatments (Kennedy et al., 2017), estimation of quantile effects (Kallus et al., 2022) and settings with limited unobserved confounding (Rambachan et al., 2024). A few works, like ours, have developed semiparametric methods for settings with interference. Several works have explored interference channeled through individuals' social networks (Emmenegger et al., 2023; Ogburn et al., 2024). Munro (2024) develops a DML theorem for settings like auctions where interference is channeled through a centralized allocation mechanism. Zhan et al. (2024) develops a DML theorem for a discrete choice model of content consumption under a neural network-based recommender system. Ballinari and Wehrli (2024) develops a DML theorem for a time-series model where a single unit is observed over time, but where observations across time obey a mixing condition and where the target estimand is an impulse response function, which measure the effect of an intervention at a particular time on the outcome at a future time. We supply an extended comparison to some key related works in Appendix B.

## 2. Modeling Shared-State Interference

We first give a high-level description of our setting. Our model begins with a set of sequentially arriving units. Each unit has covariates and a binary treatment assignment drawn iid from a joint distribution, as in canonical causal inference settings. There is also a (possibly vector-valued) observed shared state through which all interference is channeled: each arriving unit has an outcome of interest that may depend on the shared state, their covariates and their treatment assignment; then the shared state at the next time step may depend on the previous shared state, as well as the previous unit's outcome, covariates and treatment assignment. The key assumption that makes our setup tractable is that the shared state has a Markov property: that is, conditional on the shared state at the previous time step and the data of the unit that arrived in the previous time step, the shared state is independent of the previous data. We next introduce the model and notation formally.

### 2.1. Setting

**Model.** We will consider a sequence of units indexed $t = 1, 2, \ldots$, and observed up to time $T$. Each unit $t$ will have features $X_t$ drawn iid from a sample space $\mathbb{R}^{p_X}$ for finite dimension $p_X$ and a binary treatment assignment $D_t$. We will let $H_t$ on $\mathbb{R}^{p_H}$, for finite dimension $p_H$, denote the shared state. The observed outcome for each unit will be real-valued and denoted $Y_t$.

We will adopt notation so that potential outcomes are a (stochastic) function of treatment and the shared-state (i.e., $Y_t(D_t, H_t)$) similar to the exposure mapping approach of Aronow and Samii (2017). The dependence of outcomes on the shared state is the primary difference between this model and canonical iid causal inference settings, and the avenue through which interference is assumed to occur. We will assume $D_t$ is independent of all data from prior units but that it may depend on $X_t$. The observed data associated with time step $t$ will be denoted $W_t = \{X_t, D_t, H_t, Y_t\}$ on the sample space denoted $\mathcal{W}$. We will denote the full observed data $W_{1:T} = \{W_t\}_{t=1}^{T} \in \mathcal{W}^T$ and the (unknown) data distribution of $W_{1:\infty} = \{W_t\}_{t=1}^{\infty}$ as $P \in \mathcal{P}$ for some set of distributions $\mathcal{P}$. The set $\mathcal{P}$ specifies the set of probability distributions in which the true distribution may fall, and we will impose assumptions that must hold for all $P \in \mathcal{P}$.

We will assume that $H_t$ depends on the data from the previous time step (i.e., $W_{t-1}$) but is independent of $X_t$, and is conditionally independent of the data at time steps before time $t - 1$, i.e., that $H_t$ has the Markov property: For all $P \in \mathcal{P}$ and $P$-measurable $A \subseteq \mathbb{R}^{p_H}$, and $t = 1, 2, \ldots$

$$P(H_t \in A \mid W_{1:(t-1)}) = P(H_t \in A \mid W_{t-1})$$

This is assumption satisfied in settings where $H_t$ is an update to $H_{t-1}$, like when inventory at time $t$ is a function of

inventory at time $t-1$ and whether the customer at time $t-1$ made a purchase.

Our setting allows for the shared state at time $t$ to depend on the shared state at time $t-1$, allowing for dependencies over time, as long as these dependencies are exclusively mediated by the shared state. For example, the purchasing decisions of two individuals may be correlated if inventory is low during the time interval in which they arrive. The shared-state at time $t$ can also depend on the *outcome* at time $t-1$. This is different from many other models for causal inference under interference, which allow for dependencies only on other units' *treatment assignments*. (See, e.g., (Eckles et al., 2014) for further discussion of this distinction.) The dependency structure assumed by a shared-state interference setting is summarized in Figure 1.

We will also assume that the distribution of $H_t$ and $D_t$ conditional on $W_{t-1}$ is invariant of $t$. Thus, we can specify a transition probability kernel $K_H$ where, for $w \in \mathcal{W}$ and measurable event $A$ on $\mathcal{W}$ and all $t \in [T]$,

$$K_H(w, A) \stackrel{\text{def.}}{=} P\left(H_t \in A \mid W_{t-1} = w\right). \qquad (2.1)$$

There will be an initial (known) shared-state $H_0$ generated according to an arbitrary, unknown and possibly deterministic distribution $P_0$. The $t$-step kernel will be denoted $K_H^t$, where $K_H^t(w, A) = P(W_{s+t} \in A \mid W_s = w)$.

The fact that $H_t$ has the Markov property, $X_t$ is drawn iid, and $D_t$ is independent of the prior data trivially implies that $\{W_t\}_{t=1}^{\infty}$ itself is a Markov chain. Indeed, since all transition probabilities are $t$-invariant, $\{W_t\}_{t=1}^{\infty}$ is homogeneous. $\mathcal{W}$ may be uncountable, so the chain may be defined on a general (not necessarily finite) state space. We will denote the (unknown) transition probability kernel of $W_{1:\infty}$ as $K$ where $K(w, A) = P\left(W_t \in A \mid W_{t-1} = w\right)$. Note that $K$ is distinct from $K_H$; the former denotes the transition probabilities for $W_t$ and the latter only for $H_t$. As with $K_H$, the $t$-step kernel will be denoted $K^t$.

When the Markov chain has a unique stationary distribution, we will denote it $K^{\infty}$, since we will only be considering settings where, for all $w, A$ it holds $K^{\infty}(A) = \lim_{T \to \infty} K^T(w, A)$. (We drop the argument $w$ in $K^{\infty}$ since $K^{\infty}$ does not depend on the starting state.) When $K^{\infty}$ exists, we will require $K^{\infty} \in \mathcal{P}$. We will also require that $\{W_t\}_{t=1}^{\infty}$ satisfies natural Markov chain conditions.

**Assumption 2.1.** $W_{1:\infty}$ satisfies at least one of the following conditions:

(a) Geometric ergodicity and detailed balance: i.e., $W_{1:\infty}$ is a Harris ergodic Markov chain satisfying geometric mixing, and for $w, z \in \mathcal{W}$, it holds

$$K^{\infty}(dw)K(w, dz) = K^{\infty}(dz)K(z, dw). \qquad (2.2)$$

(b) $m$-dependence: i.e.,

$$W_1, \ldots, W_t \perp\!\!\!\perp W_{t+m+1}, \ldots, W_T \qquad (2.3)$$

for all $t \in [T]$.

Geometric ergodicity is a generalization (to the general state space setting) of aperiodicity and irreducibility in finite-state Markov chains; we define it formally in Appendix A. Assumption 2.1(a) implies that the chain has a unique steady state transition probability distribution and that the chain converges to that distribution from any starting state (Meyn and Tweedie, 2009, Proposition 17.1.6). Thus, writing $K^{\infty}$ in Equation (2.2) is well-defined. Intuitively, detailed balance says that, for continuous distributions, the steady state probability density around a point $w$ times the probability density of transitioning from $w$ to $z$ is equal to the same expression with the roles of $w, z$ reversed.

The $m$-dependence condition states that data observed more than $m$ steps apart are independent. Observe that any $m$-dependent sequence can be written as a Markov chain, using the last $m$ observations as the state. We will assume throughout when invoking Assumption 2.1(b) that (a finite upper bound on) $m$ is known. We refer the reader to the literature on switchback experiments for methods of estimating $m$ from data (Bojinov et al., 2022).

**Estimand.** We will assume there is a scalar-valued functional of interest $\psi^*$ which depends on the data distribution $P$; e.g., in Section 4, $\psi^*$ will be the average direct effect and in Section 5, it will be the global average treatment effect. We will assume there exists a function $\varphi^* : \mathcal{W} \to \mathbb{R}$ so that $\psi^*$ can be written as an asymptotic average over expected values of $\varphi^*$. I.e.,

$$\psi^* = \lim_{T \to \infty} \frac{1}{T} \sum_{t=1}^{T} \mathbb{E}_P\left[\varphi^*(W_t)\right].$$

We will assume that $\varphi^*$ is continuous.

**Examples of shared-state interference.** Our model is suited to describing marketplaces where interference is induced by limited supply or demand. For example, Johari et al. (2021) proposes a Markov chain model of a rental platform consisting of a sequence of arriving customers and a set of listing types, each with a fixed number of available listings. If a customer books a listing, it becomes unavailable for a time, temporarily decreasing the number of listings of the type by 1. Several other models of interference in specific markets can be well-captured by shared-state interference: for example, models of freelance labor markets (Wager and Xu, 2021) and ride-sharing markets (Li et al., 2024). Moreover, $m$-dependence is assumed in the large literature on switchback experiments (Bojinov et al., 2022).

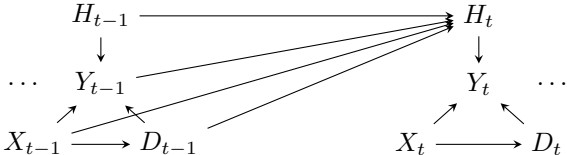

**Figure 1:** Dependency structure of shared-state interference.

## 3. DML for SSI

In this section, our main result is a double machine learning (DML) theorem for causal inference under shared state interference (SSI). Double machine learning is a meta-algorithm which allows for efficient inference with the use of expressive machine learning methods. It was proposed in Chernozhukov et al. (2018), building on a large literature in semiparametric statistics. We provide background on the method in Appendix A and give an informal overview here.

At a high level, the method consists of two steps. First, an expressive machine learning algorithm is used to approximate any nuisance parameters. Nuisance parameters often take the form of estimated conditional expectation functions (the expected outcome conditional on covariates and treatment) or propensity scores (probability of treatment conditional on covariates) which are useful for estimation of treatment effects but not of interest on their own. A predictor for the expected outcome conditional on covariates could be used to create plug-in estimators by simply taking the average of predicted outcomes under treatment versus under control. However, such plug-in estimators may converge at slower-than-parametric rates and may be very biased in finite samples. Thus, second, the estimator is constructed so as to satisfy a first-order insensitivity property, in a distributional sense that we will precisely specify later. This property can be used to show that the convergence rates of the estimator depend only on *products* of convergence rates of the ML estimators, which allows for weaker conditions on the convergence rates for any one estimator.

We note that a naive approach to inference in our setting would be to treat the shared-state as a covariate and apply DML methods as if the data were iid. However, depending on the estimand, this may yield inconsistent treatment effect or variance estimators, as we will see in our simulations in Sections 4 and 5.

An important part of any DML procedure is that the nuisance parameter estimators must be independent of the data used to construct the target estimator. (Otherwise, typical $L^2$ convergence rates are not sufficient to ensure that the nuisance estimators yield consistent target estimators.) In iid settings, this independence is usually achieved via sample splitting: the data is split into $k$ folds, the ML estimator

is trained on all but one fold and then the estimator is constructed using the held-out fold. In our setting, in the vein of Angelopoulos et al. (2023), Ballinari and Wehrli (2024) and many similar methods in statistics and machine learning, we instead assume that the nuisance estimators are generated via an auxiliary sample of data that is independent of the data to be used for inference. We make this simple alternate assumption because sample splitting would not guarantee independence between folds of the data in our setting, so predictors trained on some folds may still not be independent of the others.

For our theorem in this section, we will assume that there is a known estimator $\psi$ that depends on $W_{1:T}$ and a set of data-dependent nuisance parameters/functions $\eta \in \mathcal{S}$ for some convex set $\mathcal{S}$. We can imagine $\eta$ to be the parameters in a neural network, the coefficients on a possibly high-dimensional linear regression, or the feature splits and thresholds for trees in a random forest. We will need that $\psi$ identifies $\psi^*$; i.e., that $\mathbb{E}_P\left[\psi(W_{1:T}; \eta^*)\right] = \psi^*(P)$ for unknown true nuissances $\eta^* \in \mathcal{S}$. We will also require that $\psi$ is an empirical mean of a time-$t$ estimator $\varphi$, i.e., that

$$\psi(W_{1:T}; \eta) = T^{-1} \sum_{t=1}^{T} \varphi(W_t; \eta).$$

In Sections 4 and 5, $\psi$ will be an estimator for the average direct effect (ADE) and global average treatment effect (GATE), respectively, which we define formally under the structural models in each of those sections. The estimator will be specially constructed so as to satisfy the double-robustness property characteristic of DML methods. Much of the challenge in applying this section's main theorem involves specifying a structural model that fits a context of interest and identifying an estimator with the double-robustness properties.

We will assume access to a nuisance function estimator $\hat{\eta}$ (usually imagined to have been trained by a flexible machine learning method like a neural network or random forest). We will also require that $\hat{\eta}$ belongs to a convex set containing $\eta^*$, with high probability. This is for technical reasons that will become clear later and are common to all DML methods: the conditions required for our main theorem necessitate reasoning over convex combinations of $\hat{\eta}$ and $\eta^*$ so it is useful to define a convex set within which $\hat{\eta}$ and $\eta^*$ fall. Finally, we will require that the convex set is shrinking around $\eta^*$ asymptotically in $T$. This ensures that $\hat{\eta}$ is converging to $\eta^*$ (in an appropriate sense) as $T$ scales. Formally, for a constant $0 \leq \gamma < 1$, we will require that the nuisance function estimator $\hat{\eta}$ belongs to a convex set $S_T \subseteq \mathcal{S}$ with probability $1-\gamma$ where it is also assumed $\eta^* \in S_T$. The DML procedure for our context is summarized in Algorithm 1.

We will need several assumptions about the data generat-

---

**Algorithm 1** DML for shared-state interference

---

Train a nuisance function estimator $\hat{\eta}$ from auxiliary data $W^{\text{aux.}}$.

Construct the estimator $\hat{\psi} \stackrel{\text{def.}}{=} \psi(W_{1:T}, \hat{\eta})$.

Return $\hat{\psi}$.

---

ing process and nuisance function estimation. These assumptions are analogous to those necessary for the results in Chernozhukov et al. (2018). The first assumption, Assumption C.2, requires that $\psi$ is a smooth function of $\eta$, is Neyman orthogonal and satisfies regularity conditions. We state it formally in Assumption C.2. The second assumption, Assumption C.3, requires that the nuisance estimators obey regularity conditions and converge at appropriate rates, for all $\eta \in S_T$. We state it formally in Assumption C.3. We defer discussion and interpretation of the assumptions to the appendix. We are now ready to state our theorem, which says that the assumptions above are sufficient for efficient inference and consistent variance estimation under shared-state interference.

**Theorem 3.1.** *Under Assumptions 2.1, C.2 and C.3, with probability no less than $1 - \gamma$, Algorithm 1 returns an estimator such that*

$$\sqrt{T}\sigma^{-1}(\psi(W_{1:T}; \hat{\eta}) - \psi^*) \stackrel{d}{\to} N(0, 1) \qquad (3.1)$$

*where we define $\sigma^2 = \lim_{T \to \infty} T \cdot \text{Var}_P(\psi(W_{1:T}, \hat{\eta}))$. Moreover, when the variance of the estimator $\sigma^2$ is replaced with the variance estimator $\hat{\sigma}^2$ given in Equation (C.1), Equation (3.1) still holds.*

The proof of our Theorem 3.1 follows the pattern in that of Theorem 3.1 of Chernozhukov et al. (2018). We have simplified some of their setting for the sake of space and clarity, but our analysis in this paper could be extended to accommodate the additional generality in their theorems.

To prove the result, we must account for covariance between different terms $\varphi(W_t)$ in our analysis; Chernozhukov et al. (2018) do not have to handle any covariances since each observation is independent. To control covariance, we appeal to the combination of geometric ergodicity and detailed balance or $m$-dependence. In either case, correlation between terms goes to zero sufficiently fast that these terms do not dominate. After these covariances are handled, we may replace the iid central limit theorem used in Chernozhukov et al. (2018) with an appropriate Markov chain central limit theorem to complete the result. To prove consistency of the variance estimates, under Assumption 2.1(a) we apply a consistent variance estimation theorem from the Markov chain literature. Under Assumption 2.1(b), we show that the plug-in variance estimator is consistent by showing the plug-in estimates of each variance and covariance term are themselves consistent.

**Applying the theorem.** At a high level, instantiating Theorem 3.1 in specific settings is a matter of verifying Assumptions 2.1, C.2 and C.3. Once these are satisfied, Theorem 3.1 tells us that the DML procedure in Algorithm 1 can be used to generate asymptotically normal treatment effect estimators using the provided variance estimators.

In the next two sections, we apply Theorem 3.1 for two structural models of independent interest. In each section, we examine a specific semiparametric structural model of outcomes given covariates, treatment and the shared state and a specific causal estimand. In each case, verifying Assumptions 2.1, C.2 and C.3 amounts to ensuring all interference is channeled through a shared state, making appropriate regularity assumptions and proving the following three lemmas: first, that the estimator is Neyman orthogonal with respect to the nuisance estimators; second, that the average over $t$ of the $L^2$ norm of $\varphi(W_t; \eta) - \varphi(W_t, \eta^*)$ must converge to zero as $T \to \infty$; third, that the second Gateaux derivative of the estimator has order smaller than $\sqrt{T}$. The regularity assumptions amount to support conditions requiring a lower bound on the probabilities of each treatment assignment and requirements that nuisance estimates and parameters are bounded in probability.

# 4. Inference on Average Direct Effects

In this section, we instantiate Theorem 3.1 to estimate average direct effects in observational settings with shared-state interference. The average direct effect (ADE), informally, is the mean difference between treatment and control outcomes for each individual, keeping all other individuals' treatment assignments the same. It is also sometimes called the expected average treatment effect (Sävje et al., 2019). We first define a structural model of outcomes under interference in Section 4.1, apply our DML theorem to this setting in Section 4.2, and then validate our method for finite samples via simulations in Section 4.3.

## 4.1. Model, estimand and estimator

Informally, our structural model will consist of data generated according to a standard observational setup for units indexed $t = 1, \ldots, T$ — with an outcome $Y_t$, covariates $X_t$ and treatment assignment $D_t$ — except that we will allow for each outcome to depend on a shared state $H_t$, which is dependent on data up to time $t - 1$. The shared state, which may be vector-valued, will be the channel through which spillovers may occur in the data: outcomes are dependent on shared states and shared states may be dependent on previous outcomes, covariates, treatments and shared states. In particular cases, the shared state might represent available inventory in a market, like the number of drivers available on a ride-sharing platform, or public information, such as the popularity of songs on a music streaming platform.

We will have, for each individual $t$, potential outcomes, shared-state and treatment given according to

$$Y_t(D_t, H_t) = f^*(D_t, X_t, H_t) + \tilde{Y}_t, \qquad (4.1)$$

$$H_t(W_{t-1}) = h^*(W_{t-1}) + \tilde{H}_t, \qquad (4.2)$$

$$D_t = m^*(X_t) + \tilde{D}_t, \qquad (4.3)$$

where $Y_t(\cdot, \cdot)$ is a potential outcome depending on unit $t$'s outcome and the shared state, $H_t(\cdot)$ is the potential shared state depending on the data from time $t - 1$, $D_t$ is the treatment assignment and $X_t$ are covariates. We use $Y_t$ and $H_t$ to denote the random variables realized from the data generating process (as opposed to $Y_t(\cdot, \cdot)$ and $H_t(\cdot)$ which describe the *potential* random variables under different treatment assignments and shared state or prior data). The structural model posits that $Y_t$ is determined by a function $f^*$ plus a stochastic residual term $\tilde{Y}_t$. Similarly, $H_t$ is a function of $h^*$ and residual $\tilde{H}_t$ and $D_t$ is a function of $m^*$ and residual $\tilde{D}_t$. We will assume the residual terms $\tilde{Y}_t, \tilde{H}_t, \tilde{D}_t$ obey $\mathbb{E}_P[\tilde{Y}_t \mid D_t, X_t, H_t] = 0$, $\mathbb{E}_P[\tilde{H}_t \mid W_{t-1}] = 0$, $\mathbb{E}_P[\tilde{D}_t \mid X_t] = 0$. Thus, $m^*$ is a propensity score function since $P(D_t = 1 \mid X_t) = \mathbb{E}_P(D_t \mid X_t) = m^*(X_t)$.

By assumption, $H_t$ depends on data from prior time steps only through the observations at time $t - 1$. Thus, $H_t$ satisfies the Markov property. We also note that this model allows for $H_t$ to be stochastic, even conditional on $W_{t-1}$, through the residual term $\tilde{H}_t$. Thus, the shared state may fluctuate due to factors exogenous to the previously arriving units' behavior. We also assume that $H_t$ satisfies either Assumption 2.1(a) or Assumption 2.1(b). This implies that the whole chain, consisting of observations $\{W_t\}_{t=1}^T$ satisfies Assumption 2.1.

Our estimand in this section will be the average direct effect (ADE). It is defined:

$$\psi^* \stackrel{\text{def.}}{=} \frac{1}{T} \sum_{t=1}^T \mathbb{E}_P \left[ Y_t(1, H_t) - Y_t(0, H_t) \right] \qquad (4.4)$$

where the expectation is taken over $W_{1:T}$. In other words, this is the average effect on each individual of changing their treatment from control to treatment, without altering the distribution of others' treatments. (Note that the expectation $\mathbb{E}_P$ marginalizes over values of $H_t$, so the estimand is not conditional on realizations of $H_t$.)

Recall from Section 3 that, in our setting the estimator $\psi(W_{1:T}; \eta)$ is the sample average of functions $\varphi(W_t; \eta)$. In this section (and the next), the functional form of the time-$t$ estimator $\varphi$ can be written as the sum of two components: a plug-in term and a debiasing term. For convenience, we will denote these $\varphi^{\text{pi}}$ and $\varphi^{\text{db}}$; i.e., $\varphi(W_t; \eta) = \varphi^{\text{pi}}(W_t; \eta) + \varphi^{\text{db}}(W_t; \eta)$. These will be defined in this section as

$$\varphi^{\text{pi}}(W_t, \eta) \stackrel{\text{def.}}{=} f(1, X_t, H_t) - f(0, X_t, H_t),$$

$$\varphi^{\text{db}}(W_t, \eta) \stackrel{\text{def.}}{=} \left( \frac{D_t}{m(X_t)} - \frac{1 - D_t}{1 - m(X_t)} \right) \qquad (4.5)$$
$$\cdot (Y_t - f(D_t, X_t, H_t)),$$

where the nuisance estimators are $\eta = (f, m)$. The function $\varphi$ mirrors the augmented inverse probability weighting (AIPW) estimator (Robins et al., 1994) that is standard in the semiparametric inference literature (see, e.g., Wager (2024) for background and history), with the exception that our outcome nuisance estimator $f$ takes the shared state $H_t$ as an argument. One way to interpret our estimator is as a covariate adjustment for the shared state: omitting the shared-state at a given time may result in a form of confounding between units' outcomes. However, it is not sufficient to treat $H_t$ as a covariate and apply DML methods as if the data were iid, since there is dependence between units which needs to be accounted for. We will see the contrast between our method and an approach that treats the shared state as if they were iid covariates in our simulations, and we note that, in Section 5, treating the shared state as if they are covariates will not in general yield a consistent treatment effect estimator.

## 4.2. Estimation and inference

Our main result in this section requires two assumptions beyond the Markov chain assumption. We defer formal definitions and further discussion of these assumptions to the appendix and discuss them briefly next.

Assumption D.1 imposes regularity conditions which ensure that both the true data generating process and estimated nuisance functions satisfy support conditions on the probability of treatment and that nuissance functions and error terms are bounded in $L^q$ norm. Assumption D.2 requires that the nuisance estimators $\hat{m}$ and $\hat{f}$ converge in $L^2$ to their true values $m^*$ and $f^*$ and that the product of their convergence rates vanishes at $o_P(T^{-1/2})$ rates.

The required rates for Assumption D.2 can be achieved if, for example, when averaging over time steps, $\|(m - m^*)(X_t)\|_{L^2(P)} = o(T^{-1/4})$ and $\|(f - f^*)(D_t, X_t, H_t)\|_{L^2(P)} = o(T^{-1/4})$. On the other hand, if the propensity score function $m^*$ is known (if, e.g., it is determined via a randomized experiment controlled by the researcher), then $\|(f - f^*)(D_t, X_t, H_t)\|_{L^2(P)}$ can converge at an arbitrarily slow rate and still yield an efficient estimator.

There is a rich literature giving learning rates for machine learning estimators trained on dependent data. A key implication of our Markov chain assumptions is that the data are $\rho$-mixing, which intuitively means that observations

that are far apart have diminishing correlations (see, e.g., (Bradley, 2007) for formal definitions of mixing conditions). Learning rates for mixing sequences have been proved for neural networks (Ma and Safikhani, 2022), random forests (Goehry, 2020) and several other machine learning methods (Irle, 1997; Steinwart et al., 2009; Lozano et al., 2014; Wong et al., 2020). See (Ballinari and Wehrli, 2024) for further discussion.

With these two assumption in hand, we are ready to state our efficient inference result for this section.

**Theorem 4.1.** *For the model defined in Section 4.1 and the estimand $\psi$ and estimator $\psi^*$ defined in Equations* (4.4) *and* (4.5) *respectively, under Assumptions D.1 and D.2, then, with probability no less than $1 - \gamma$, it holds*

$$\sqrt{T}\hat{\sigma}^{-1}(\psi(W_{1:T};\hat{\eta}) - \psi^*) \xrightarrow{d} N(0,1)$$

*where $\hat{\sigma}$ is defined in Equation* (C.1).

Theorem 4.1 instantiates Theorem 3.1 for the model defined in Section 4.1. It replaces the general assumptions required for Theorem 3.1 with specific regularity and rates assumptions in Assumptions D.1 and D.2.

### 4.3. Simulations and estimator comparison

Next, we validate our results through simulations demonstrating the performance of the double machine learning estimator versus naive alternate estimators.

**Setup for treatment effect estimator comparisons.** We compare our estimator, which we will sometimes denote by DML4SSI, to several benchmark estimators: The naive plug-in estimator $\psi^{\mathrm{pi}}$, defined formally in Equation (F.1), simply imputes the conditional expectation for each treatment assignment using the estimated conditional expectation function $\hat{f}$. Plug-in estimators constructed using machine learning models will converge at slower-than-$\sqrt{T}$ rates, and in finite samples may be very biased. The naive difference-in-means Horvitz-Thompson (HT) estimator $\psi^{\mathrm{HT}}$, defined formally in Equation (F.2), takes a difference in means between treatment and control units. The naive HT estimator does not account for the shared state and thus may be inconsistent and very biased in finite samples. A naive DML estimator (DML-N) treats the data as if it were iid using the AIPW estimator and is defined in Equation (F.3).

In our simulations, we generated data according to a smooth function of both $D_t, X_t$ and of $H_t$. For all machine learning predictors $\hat{f}$, $\hat{m}$ and $f'$, we used random forests with default parameters trained on auxiliary data of size $T$ sampled independently of the data used for inference. Then, we computed each of the estimators $\psi(W_{1:T};\hat{\eta}), \psi^{\mathrm{HT}}(W_{1:T};\hat{\eta})$, $\psi^{\mathrm{pi}}(W_{1:T};\hat{\eta})$ and $\psi^{\mathrm{DML-N}}(W_{1:T};\eta')$ and created a density plot for the bias of the resulting estimates in Fig-

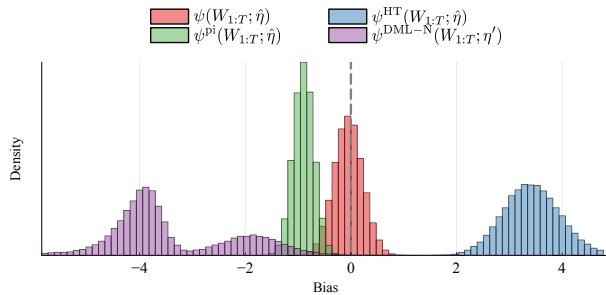

**Figure 2:** Estimates of the average direct effect with our double machine learning estimator versus naive Horvitz-Thompson, naive plug-in and naive DML estimators.

ure 2. Full details of the simulations are available in Appendix F and the code used to generate them is available at `github.com/johnchrishays/dml4ssi`.

**Results for treatment effect estimator comparisons.** In Figure 2, we observe that all the distributions of each of the naive estimators are substantially biased. By contrast, the DML estimator is centered around the true treatment effect. The bias in the plug-in estimator occurs despite the smoothness of $f^*$ and $m^*$ and the fact that the data is low dimensional. The bias in the HT and DML-N estimators comes from the fact that they do not account for the effect of the shared state and thus will in general be inconsistent. The direction and magnitude of the bias for each naive estimator is idiosyncratic to our synthetic data generating process and machine learners; in other settings the sign of the biases may be reversed, and the relative performance of different naive estimators may change.

## 5. Regression adjustments in switchback experiments

In this section, we turn our attention to estimating the global average treatment effect (GATE) using switchback experiments. The GATE, as we will define formally shortly, is the difference in an the mean outcome of interest when all units are assigned to treatment versus when all units are assigned to control. Switchback experiments, which were formalized in Bojinov et al. (2022), allow for valid inference in settings where interference is $m$-dependent.

Switchback experiments are implemented by assigning treatments in sequential blocks so that $\ell > m$ units in a row receive the same (randomized) treatment. For example, for a block of $\ell$ sequential units, the experimenter may use a Bernoulli random draw to assign all units in the block to treatment or to control. Intuitively, switchback experiments account for interference by arranging for the observation of some units where all $m$ previous units received the same

treatment assignment as the current unit. Our framework incorporates doubly robust regression adjustments that allow for lower variance estimates in settings where covariates and shared-state variable exert significant influence on outcomes (or, equivalently, in cases where treatment effect sizes are small). The structure of the remainder of this section is the same is in Section 4: we define our setting in Section 5.1, state an efficient inference result in Section 5.2, and validate our results via simulations in Section 5.3.

## 5.1. Model, estimand and estimator

Our model in this section is similar to that of Section 4.1: for $t = 1, \ldots, T$, we have an outcome $Y_t$ that depends on a shared state $H_t$, covariates $X_t$ and treatment assignment $D_t$. This section is different from the previous in that we require $H_t$ to consist of the last $m$ observations of treatment assignments and covariates. This allows for satisfying the $m$-dependence condition required for switchback experiments to yield unbiased estimates of treatment effects (Bojinov et al., 2022). Also, since we are considering switchback experiments, we assume $D_{1:T}$ to come from a known distribution determined by the experimenter.

Formally, we will analyze the following potential outcomes model on a population indexed $t = 1, 2, \ldots$. For a constant $m \in \mathbb{N}$ and for each individual $t$ there is some binary treatment of interest $D_t \in \{0, 1\}$, iid covariates $X_t \in \mathbb{R}^d$, shared state $H_t \in \mathbb{R}^{d_H}$ and outcome $Y_t \in \mathbb{R}$ given by

$$Y_t(D_t, H_t) = f^*(D_t, X_t, H_t) + \tilde{Y}_t, \tag{5.1}$$

$$H_t(D_{(t-m):(t-1)}) = \texttt{vec}(D_{(t-m):(t-1)}, X_{(t-m):(t-1)}) \tag{5.2}$$

$$D_{1:T} \sim \mathcal{D} \tag{5.3}$$

where $\mathcal{D}$ is a distribution determined by the experimenter. We will drop the arguments to $H_t$ when $D_{(t-m):(t-1)}$ are the observed treatment assignments. The residual terms $\tilde{Y}_t$ will be assumed to be mean zero and obey $\mathbb{E}_P[\tilde{Y}_t \mid D_t, X_t, H_t] = 0$. Notice that the above implies that shared states are $m$-dependent as defined in Equation (2.3). For an index set $\mathbf{t} = (s : t)$, $s, t \in [T]$ and boolean vector $\mathbf{b} = \{0, 1\}^{|\mathbf{t}|}$, define $\pi^*(\mathbf{t}, \mathbf{b}) = \mathbb{P}[D_{\mathbf{t}} = \mathbf{b}]$ to be the probability that the subvector $D_{\mathbf{t}}$ is equal to $\mathbf{b}$. The estimand will be the GATE, defined as

$$\psi^* = \frac{1}{T} \sum_{t=1}^{T} \mathbb{E}_P[Y_t(1, H_t(\mathbf{1})) - Y_t(0, H_t(\mathbf{0}))]. \tag{5.4}$$

As in the previous section, our estimator $\psi$ will be defined by the sample average of the sum of a plug-in term $\varphi^{\mathrm{pi}}(W_t, \eta)$ and a debiasing term $\varphi^{\mathrm{db}}(W_t, \eta)$. We define these terms for this section as follows:

$$\varphi^{\mathrm{pi}}(W_t, \eta) \stackrel{\text{def.}}{=} f(1, X_t, H_t(\mathbf{1})) - f(0, X_t, H_t(\mathbf{0})),$$

$$\varphi^{\mathrm{db}}(W_t, \eta) \stackrel{\text{def.}}{=} \left( \frac{\mathbb{1}\{D_{(t-m):t} = \mathbf{1}\}}{\pi^*((t-m):t, \mathbf{1})} - \frac{\mathbb{1}\{D_{(t-m):t} = \mathbf{0}\}}{\pi^*((t-m):t, \mathbf{0})} \right)$$
$$\cdot (Y_t - f(D_t, X_t, H_t)). \tag{5.5}$$

Note that the estimator has a form similar to the AIPW estimator of the previous section with a few differences. First, in the plug-in component of the estimator, we plug in the all treatment or all control vector for treatment assignments. Second, in the debiasing component of the estimator, we substitute in indicator functions for the $m$ past treatment assignments being equal to the current units' treatment assignment. These ensure that the estimator identifies the effect of all treatment versus all control.

We will assume $\pi^*((t-m):t, \mathbf{1}), \pi^*((t-m):t, \mathbf{0}) > \zeta$ for all $t$. This is not always the way that switchback experiments are defined: in some contexts, the switching points may be deterministic, which would mean that any $t$ less than $m$ time steps after a (deterministic) switch point would have $\pi^*((t-m):t, \mathbf{1}) = \pi^*((t-m):t, \mathbf{0}) = 0$. One way to achieve the constraint is to pick a switching length $\ell > m$ and then pick a first switch uniformly at random from the first $\ell$ time steps. Then for each block of constant treatment assignments, Bernoulli randomize. This will give $P(D_{(t-m):t} = \mathbf{1}) = P(D_{(t-m):t} = \mathbf{0}) \geq (\ell - m)/2\ell$.

## 5.2. Estimation and inference

The assumptions necessary for efficient inference in switchback experiments under $m$-dependence are analogous to those in Section 4 and are deferred to Appendix E. The only crucial difference is that, since treatment assignments are randomized in an experiment, propensity scores are known. This means we do not need to impose rate conditions on the convergence of $f$ to $f^*$. We now state our estimation result for the GATE under our model.

**Theorem 5.1.** *For the model defined in Section 5.1, and the estimand and estimator defined in Equations* (5.4) *and* (5.5), *under Assumptions E.1 and E.2, then with probability no less than* $1 - \gamma$, *it holds*

$$\sqrt{T}\hat{\sigma}^{-1}(\psi(W_{1:T}; \hat{\eta}) - \psi^*) \stackrel{d}{\to} N(0, 1),$$

*where $\hat{\sigma}$ is as defined in Equation* (C.1).

## 5.3. Simulations and estimator comparison

We next validate the DML method for estimation of the GATE in switchback experiments.

**Setting.** To guarantee $m$-dependence, the data generating process used for the plots in this section is different from that in Section 4.3. In particular, the data generating process for $H_t$ in Section 4.3 allowed for dependence on prior shared

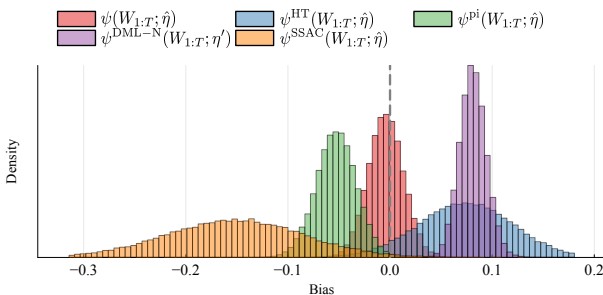

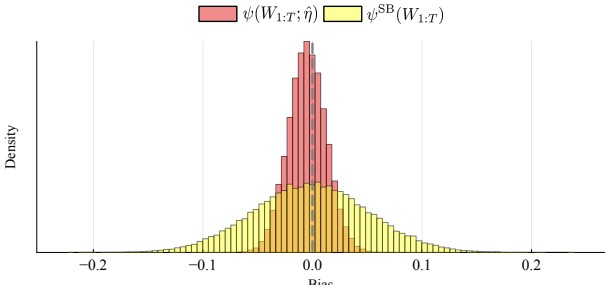

**Figure 3:** Estimates of the global average treatment effect in switchback experiments with our double machine learning estimator versus naive estimators.

**Figure 4:** Estimates of the GATE in switchback experiments with the double machine learning estimator versus the switchback Horvitz-Thompson estimator. The DML estimator has substantially lower variance.

states, which allows for long-range dependencies between observations over time. In this section, $H_t$ is a function only of the last $m$ observations of treatments $D_{t-m:t-1}$ and covariates $X_{t-m:t-1}$. The setup is otherwise similar to that in Section 4.3: $f^*$ is a smooth function of $D_t, X_t, H_t$ and we generate the predictor $\hat{f}$ using a random forest with default parameters. The switching length is chosen arbitrarily $\ell = 2m$. Full details of the simulations are deferred to Appendix F. We compare the performance of our estimator against naive estimators in Figure 3 and against an unbiased estimator for treatment effects in switchback experiments in Figure 4.

**Treatment effect estimator comparisons.** In Figure 3, we compare our DML estimator to naive estimators. As in Section 4.3, we compare our estimator to a naive Horvitz-Thompson estimator, a plug-in estimator and a naive DML estimator. Each estimator is defined the same way as in Section 4.3, except that we plug in the known true propensity score $m^*$ instead of the estimated propensity score $\hat{m}$ since $m^*$ is known in an experiment. Additionally, we show the shared state as covariates estimator defined in Equation (F.5). $\psi^{\text{SSAC}}$ is the estimator obtained by treating the shared state as iid covariates.

Note that the naive estimators are biased away from the true GATE, while the DML estimator is approximately centered at the true effect. The SSAC estimator in this section is biased, intuitively, because it *controls for* the spillover effects between units, rather than *incorporating* them into the effect estimate: the global average treatment effect includes both the direct and indirect effects of treatment.

In Figure 4, we compare our estimator with an unbiased switchback estimator proposed in Bojinov et al. (2022), which is defined in Equation (F.6). Note that both the DML and switchback HT estimator distributions are centered at the true effect, but the DML estimator has substantially lower variance. This is due to the fact that some of the variation in outcomes is explained by the shared state and

covariates (and is learned by expressive machine learning models), and the fact that the DML estimator uses all of the data while the switchback HT estimator drops the first $m$ observations after any switch. We leave more detailed exploration of the performance of the estimand (compared to (1) the Horvitz-Thompson-style switchback estimator and (2) across different choices of $\ell$) for future work.

## 6. Discussion

In this paper, we formally define shared-state interference and develop methods for efficient estimation of treatment effects using double machine learning. Shared-state interference occurs when spillovers in a system are channeled through a low-dimensional statistic like prices, inventory or information. Our motivating examples of shared-state interference are marketplaces and recommender systems, but there are many contexts where researchers and practitioners wish to measure treatment effects in the presence of shared-state interference. Our method allows for the use of expressive machine learning estimation of nuisance parameters while achieving $\sqrt{T}$-rates. We also provide a consistent variance estimator.

In future work, it would be valuable to explore a number of theoretical and empirical questions around shared-state interference. Theoretically, it would be interesting to explore what other estimands (e.g., the average indirect effect as in Munro et al. (2023) or time-discounting as in Farias et al. (2023)). It would also be valuable to characterize alternate forms of interference that have a similar structure: for example, in some situations, a shared state may influence treatment assignments (which we assume does not occur in our structural models). Empirically, it would be interesting to apply our methods to markets or information systems that exhibit shared-state interference. We hope that this work inspires further investigations of tractable interference structures in semiparametric inference.

## Impact Statement

This paper presents work whose goal is to advance the methods for measuring treatment effects in settings like recommender systems and markets where units interact via shared social signals or information. There are many potential societal consequences of our work, none which we feel must be specifically highlighted here.

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

# A. Background

**Additional notation.** For a vector-valued random variable $V$, when we write $V = O_p(\cdot)$, we mean that the relation holds element-wise. For a sample space $\Omega$, let $\mathcal{B}(\Omega)$ be the corresponding Borel set.

**Geometrically ergodic Markov chains.** A geometrically ergodic Markov chain is a Harris ergodic chain that satisfies a geometric mixing rate property. A Harris ergodic chain is aperiodic, $\mu$-irreducible where $\mu$ is a "maximal" irreducibility measure (see, e.g., (Meyn and Tweedie, 2009, Theorem 4.0.1)) and positive Harris recurrent. We define each of these properties next.

First, we review aperiodicity for our setting. The period of the chain $k$ is defined in our context as the largest sequence of disjoint measurable sets $A_0, \ldots, A_{k-1}$ such that the chain cycles between the sets with probability one: i.e., that $P(W_t \in A_{(i+1)\%k} \mid W_{t-1} \in A_i) = 1$ for all $i \in \{0, \ldots, k-1\}$. An aperiodic chain is one where the period is 1. A chain is $\mu$-*irreducible* for some measure $\mu$, if, for all measurable $A \subseteq \mathcal{W}$ such that $\mu(A) > 0$, it holds for all $w \in A$ and $i \in [T]$

$$P(\exists t \in \mathbb{N} \ : \ W_{t+i} \in A \mid W_i = w) > 0.$$

This definition of irreducibility is a generalization of irreducibility in finite-state Markov chains. *Harris recurrent* means that for all $\mu$-measurable $A \subseteq \mathcal{W}$ with $\mu(A) > 0$ and $w \in A$, it holds,

$$P(\{W_t\}_{t=1}^\infty \in A \text{ i.o.} \mid W_1 = w) = 1,$$

where i.o. stands for infinitely often:

$$\{\{W_t\}_{t=1}^\infty \in A \text{ i.o.}\} \stackrel{\text{def.}}{=} \bigcap_{s=1}^\infty \bigcup_{t=s}^\infty \{W_t \in A\}.$$

Harris recurrence is a generalization of recurrence in finite-state Markov chains. A chain is *positive* if there exists a measure $\tau$ such that, for all $\tau$-measurable $A$,

$$\tau(A) = \int_{\mathcal{W}} K(w, A)\tau(dw),$$

i.e., that there exists an invariant measure.

*Detailed balance* is defined in Equation (2.2). The property means, for continuous state spaces, that the steady state probability density of each state times the probability density of the transition to another state is equal to the steady state probability density times the density of transition to the first state.

A Harris ergodic Markov chain is *geometrically ergodic* if there exists a function $M \ : \ \mathcal{W} \to \mathbb{R}_{>0}$ and constant $\theta \in (0, 1)$ such that, for all $w \in \mathcal{W}$,

$$\left\| P^T(w, \cdot) - P_\infty(\cdot) \right\|_{\text{TV}} \leq M(w)\theta^T,$$

where $\|\cdot\|_{\text{TV}}$ is the total variation distance norm defined for a signed measure $\mu$ as

$$\|\mu\|_{\text{TV}} = \sup_{A \in \mathcal{B}(\mathcal{W})} \mu(A) - \inf_{A \in \mathcal{B}(\mathcal{W})} \mu(A).$$

**Double machine learning (DML).** Double machine learning is a meta-algorithm for semiparametric inference, introduced and analyzed in Chernozhukov et al. (2018). It allows for the use of expressive machine learning models even when those models may not converge to a true data generating process at parametric rates. The algorithm consists of two conceptual components: sample splitting and a bias correction in the estimator. The first component is the construction of nuisance function approximators from data that is independent of the observed data it will be applied to, typically achieved by sample splitting. The nuisance estimators can be constructed using expressive machine learning methods that may produce biased predictions and converge at slower-than-parametric rates. The second component is the use of a bias correction term added to a plug-in estimator which ensures that errors in the nuisance function approximators depend only on second-order terms. We use the following definitions of the Gateaux derivative and Neyman orthogonality in our results.

**Definition A.1** (Gateaux derivative). For a convex set of functions $\mathcal{S}$ and function $\eta \in \mathcal{S}$, the *Gateaux derivative* of a map $f : \mathcal{S} \to \mathbb{R}$ with respect to $\eta, \eta^* \in S$ is defined as the quantity

$$\frac{\partial}{\partial r} f(\eta^* + r(\eta - \eta^*)), \quad r \in [0, 1].$$

**Definition A.2** (Neyman orthogonality). For a distribution $P$ on data $W$, the function $\psi$ is said to be *Neyman orthogonal with respect to $S$* for $\eta^* \in \mathcal{S}$ and all $\eta \in \mathcal{S}$ the Gateaux derivative of $\eta \mapsto \mathbb{E}_P[\psi(W; \eta)]$ with respect to $\eta, \eta^*$ vanishes at $r = 0$:

$$\frac{\partial}{\partial r} f(\eta^* + r(\eta - \eta^*)) \bigg|_{r=0} = 0.$$

# B. Additional Related Work

Here we provide a detailed comparison of several prior works that are particularly closely related to our paper.

*Comparison to Chernozhukov et al. (2018).* We prove a theorem analogous to their Theorem 3.1 for the case of dependent data. We make two changes to their setup. The first is that we will not assume each individual's treatment assignment, covariates and outcomes $(D, X, Y)$ are drawn iid from a joint distribution. Instead, we will assume that covariates and treatment assignments are drawn iid, that units arrive sequentially, and that outcomes are a function of covariates and a shared state possibly depending on the treatment assignments, covariates and outcomes of previous arrivals. Second, we assume, in the style of Angelopoulos et al. (2023), there is access to an auxilliary sample on which to train the nuisance parameters. We will assume that the size of the auxilliary sample scales appropriately so that the nuisance functions converge in $L^2(P)$ at appropriate rates and are independent of the data. In Chernozhukov et al. (2018), independence is achieved through sample splitting of the iid data. In our context, where outcomes are not drawn iid, the auxiliary training data serves the role that sample splitting does in the double ML for iid data. Notationally, our score function $\psi$ is defined differently than theirs: ours must be a function of all of the data $W_{1:T}$ whereas theirs may be a function of a single draw $W$ from the iid data distribution.

*Comparison to Emmenegger et al. (2023).* As in their main results, we derive efficient inference results for the average direct effect. In their appendix, they also provide results that are analogous to ours for estimation of the global average treatment effect. Our model can be written as a network interference model by constructing a network where each node affects the potential outcomes of future nodes. However, they prove results about network interference under a sparsity assumption, and our network interference model is not sparse (since there would be a directed edge from each node $t$ to each future node $s > t$). Instead, our framework makes the interference structure tractable by mediating interference through shared states. Thus, our work allows for potentially long-range dependencies between observations that are disallowed by the sparsity assumptions in the network interference literature. Additionally, our model allows for dependencies between units' *outcomes* and future units' outcomes, which is disallowed in their model (and the vast majority of work in the network interference literature).

*Comparison to Ballinari and Wehrli (2024).* Their DML results, like ours are about a time series model where units are observed in a sequence over time. They instantiate a structural model similar to our approach in Sections 4 and 5, and their efficient influence results, like ours, go through mixing conditions. The estimand in their structural model is a impulse response function, which measures the effect of an intervention at time $t$ on outcomes at time $t + h$ for some known $h$, whereas ours is the average direct effect and global average treatment effect. Our structural models are also different, since we require all interference across time to occur through shared-state variables, whereas they allow for generic mixing across time.

*Comparison to Munro (2024).* This paper considers semiparametric estimation of the global average treatment effect in settings where outcomes are determined by a centralized mechanism like an auction, rather than a context- and mechanism-agnostic setting like outs. Informally, their key assumptions require that the mechanism allocates goods according to a known function of individuals' bids and market-clearing cut-offs. They also require individuals' "bids" to the mechanism obey the stable unit treatment value assumption (SUTVA). Their work establishes estimation of the global average treatment effect in observational settings, whereas we provide methods for estimating the average direct effect in observational settings and the global average treatment effect in experimental settings. Their model considers simultaneous arrivals of units to the market, whereas ours considers sequential arrivals.

*Comparison to Zhan et al. (2024).* We prove a theorem analogous to their main theorem, except that our results go through Markov chains and theirs go through martingales. We also do not make functional form assumptions about the relationship between outcomes and the shared state, which in their case is the set of recommendations chosen by the platform (they use a discrete choice model). We also do not require that the shared state be computed by a neural network, as they do.

*Connections to the network interference literature.* Our model can be interpreted as a network with a particular structure (where each unit affects the outcomes of future units), but our model allows for dependencies between outcomes of each unit and the outcomes of subsequent units. Typically, in the network interference literature, all dependencies are channeled through treatment assignments, rather than outcomes, since outcomes in their models are typically not sequentially determined (see, e.g. Aronow et al. (2020) for an overview). Additionally, methods in network interference often make a sparsity assumption on the interference graph (like those in Emmenegger et al. (2023)), whereas our interference network can be dense. However, since correlation between units dissipates over time, it would be interesting to explore whether models of approximate neighborhood interference (Leung, 2022) or approximate local interference (Chin, 2019).

## C. Further details and proofs for Section 3

In this section, we prove our main theorem in Section 3. First, we state our consistent variance estimator.

$$
\hat{\sigma}^2 \stackrel{\text{def.}}{=} \begin{cases} \dfrac{1}{T_2(T_1-1)} \displaystyle\sum_{t=0}^{T_1-1} \left( \displaystyle\sum_{s=tT_2+1}^{(t+1)T_2} \varphi(W_s;\hat{\eta}) - \psi(W_{1:T};\hat{\eta}) \right)^2 & \text{under Assumption 2.1(a)} \\[3ex] \dfrac{1}{T} \displaystyle\sum_{t=1}^{T} (\varphi(W_t;\hat{\eta}) - \psi(W_{1:T};\hat{\eta}))^2 + 2\displaystyle\sum_{i=1}^{m} (\varphi(W_t;\hat{\eta}) - \psi(W_{1:T};\hat{\eta}))(\varphi(W_{t-i};\hat{\eta}) - \psi(W_{1:T};\hat{\eta})) \\ & \text{under Assumption 2.1(b)} \end{cases}
\tag{C.1}
$$

where, for some $\theta > (1+\delta/2)^{-1}$, we define $T_2 = \lfloor T^\theta \rfloor$ and $T_1 = \lfloor T/T_2 \rfloor$.

*Remark* C.1. We briefly comment on the fact that our variance estimators under Assumption 2.1(a) and under Assumption 2.1(b) are different. Under Assumption 2.1(a), our consistent variance estimator is not a plug-in estimator. In fact, plug-in variance estimators in Markov chains are not in general consistent (Jones et al., 2006). On the other hand, for Assumption 2.1(b), the plug-in variance estimator *is* consistent.

We next state the formal assumptions sufficient for the theorem. Assumption C.2(a) requires that second Gateaux derivatives of $\psi$ with respect to $\eta \in S_t$ exist and are continuous. The Gateaux derivative is defined in Definition A.1 and, informally, is equivalent to the usual derivative with respect to $r \in (0,1)$ for $\psi(W_{1:T}; r\eta^* + (1-r)\eta)$ for $\eta \in S_T$. Assumption C.2(b) requires Neyman orthogonality, which is defined in Definition A.2 and, informally, is the requirement that the Gateaux derivative above, evaluated at $r = 0$, is zero. In applications, using our theorem in this section will require carefully designing estimators so that these conditions hold.

**Assumption C.2** (Smoothness and Neyman orthogonality conditions). For all $T \geq 1$, the following conditions hold for all $P \in \mathcal{P}$:

(a) The map $\eta \mapsto \mathbb{E}_P[\psi(W_{1:T};\eta)]$ is twice continuously Gateaux-differentiable on $S_T$ around $\eta^*$.

(b) The estimator $\psi$ is Neyman orthogonal with respect to the nuisance realization set $S_T$ around $\eta^*$.

We next state regularity and convergence rate assumptions for our main theorem. Equation (C.2) requires that the $L^q$ norm of $\varphi(W_t, \eta)$ must be bounded for all $t \in [T]$ for $q > 4$. Equation (C.3) requires the average over $t$ of the $L^2$ norm of $\varphi(W_t; \eta) - \varphi(W_t, \eta^*)$ must converge to zero as $T \to \infty$. Finally, Equation (C.4) requires the second Gateaux derivative of $\psi$ with respect to $\eta$ must go to zero at faster-than-$\sqrt{T}$ rates. These are analogous to standard regularity and convergence rate conditions for DML theorems (Chernozhukov et al., 2018).

**Assumption C.3** (Regularity and convergence rate conditions). There exists constants $\gamma, \delta, C > 0$ and $T_0 \geq 1$ such that for all $T > T_0$, the nuisance function estimator $\hat{\eta}(W^{\text{aux}\cdot})$ belongs to a realization set $S_T$ with probability at least $1 - \gamma$ where $S_T$ contains $\eta^*$ and satisfies the following conditions for all $P \in \mathcal{P}$:

$$
\sup_{\eta \in S_T} \|\varphi(W_t;\eta)\|_{L^{4+\delta}(P)} < C, \ \forall t \in T
\tag{C.2}
$$

$$\sup_{\eta \in S_T} T^{-1} \sum_{t=1}^{T} \|\varphi(W_t; \eta) - \varphi(W_t; \eta^*)\|_{L^2(P)} = o(1), \tag{C.3}$$

$$\sup_{r \in (0,1), \eta \in S_T} \left| \partial_r^{(2)} \mathbb{E}_P \left[ \psi(W_{1:T}; \eta^* + r(\eta - \eta^*)) \right] \right| = o(T^{-1/2}) \tag{C.4}$$

We next provide an overview of the proof and prove the main theorem. At a high level, the two parts of the theorem are $\sqrt{T}$-asymptotic normality of the treatment effect estimator and consistency of the variance estimator. Here we overview the proofs of each of these parts.

For $\sqrt{T}$-asymptotic normality of the treatment effect estimator, we make the simple observation that the difference between the treatment effect estimator $\psi(W_{1:T}; \hat{\eta})$ and the estimand $\psi^*$ can be broken down into the sum of three differences. (This is an observation made in Kennedy (2023, page 19, equation 10) and is standard in proofs of DML theorems.) The three components in the sum are: (1) The difference between an estimator where the true nuisances are known (sometimes called an oracle estimator) and the estimand (Equation (C.5)), (2) the deviation from its mean of the difference between the oracle estimator and our DML estimator where the nuisances are not known (Equation (C.6)), and (3) the expected difference between the oracle and plugin estimators. Then, the first term is shown to be $\sqrt{T}$-asymptotically normal in Lemma C.6, and the second two terms can be shown to go to zero in probability at faster than $\sqrt{T}$-rates in Lemmas C.7 and C.9, respectively. Together, this implies $\psi(W_{1:T}; \hat{\eta})$ approaches its mean at $\sqrt{T}$ rates and is asymptotically normal. The consistency of the variance estimator is given by Lemma C.10. We describe the high-level intuition for each of the lemmas before their proofs.

*Proof Theorem 3.1.* Observe:

$$\psi(W_{1:T}; \hat{\eta}) - \psi^* \le \psi(W_{1:T}; \eta^*) - \psi^* + |\psi(W_{1:T}; \hat{\eta})) - \psi(W_{1:T}; \eta^*))|$$

$$\le \psi(W_{1:T}; \eta^*) - \psi^* \tag{C.5}$$

$$+ \left| \psi(W_{1:T}; \hat{\eta}) - \psi(W_{1:T}; \eta^*) - \mathbb{E}_P[\psi(W_{1:T}; \hat{\eta}) - \psi(W_{1:T}; \eta^*) \mid W^{\text{aux.}}] \right| \tag{C.6}$$

$$+ \left| \mathbb{E}_P \left[ \psi(W_{1:T}; \hat{\eta}) - \psi(W_{1:T}; \eta^*) \mid W^{\text{aux.}} \right] \right| \tag{C.7}$$

from applying the triangle inequality twice. Then applying Lemma C.7 and Lemma C.9 implies

$$\sqrt{T} \sigma^{-1} \left( \psi(W_{1:T}; \hat{\eta}) - \psi^* \right) = \sqrt{T} \sigma^{-1} (\psi(W_{1:T}; \eta^*) - \psi^*) + o_P(1)$$

Applying Lemma C.6, we observe

$$\sqrt{T} \sigma^{-1} (\psi(W_{1:T}; \eta^*) - \psi^*) \xrightarrow{d} N(0, 1).$$

which, with, e.g., (van der Vaart, 2000) Theorem 2.7(iv), implies the first statement in the result. Lastly, Lemma C.10 proves $\hat{\sigma}_T^2 \xrightarrow{p} \sigma^2$, and applying Slutsky's theorem completes the proof. $\square$

In the proof of Lemma C.6, we will use the following two central limit theorems. The first is for geometrically ergodic Markov chains on general state spaces, and the second is for $m$-dependent sequences. Theorem C.4 generalizes standard central limit theorems for Markov chains in finite state spaces, where the analogous requirements are that the chain is irreducible and aperiodic. See Appendix A for further description of geometrically ergodic Markov chains. Theorem C.5 states a central limit theorem for $m$-dependent sequences.

**Theorem C.4** (Theorem 2, (Chan and Geyer, 1994)). *Suppose that a sequence of random variables $\{A_t\}_{t=1}^{\infty}$ is a geometrically ergodic Markov chain with stationary distribution $\pi$. Also, suppose that there exists a constant $\delta > 0$ so that a measurable function $f$ satisfies $\|f\|_{L^{2+\delta}(\pi)} < \infty$. Then it holds*

$$\frac{1}{\sqrt{T} \sigma} \left( \sum_{t=1}^{\infty} f(A_t) - \mathbb{E}_\pi[f] \right) \xrightarrow{d} N(0, 1),$$

*where we define*

$$\sigma^2 \overset{\text{def.}}{=} \text{Var}_\pi(f(A_1)) + 2 \sum_{t=1}^{\infty} \text{Cov}_\pi(f(A_1), f(A_t)).$$

**Theorem C.5** (Theorem 1, (Hoeffding and Robbins, 1948)). *For a sequence of $m$-dependent random variables $A_1, A_2, \cdots$ if there exists constant $C > 0$, satisfying, for all $t = 1, 2, \ldots, \mathbb{E}_P |A_t|^3 \leq C$. Then,*

$$T^{-1/2} \sigma^{-1} \sum_{t=1}^{T} (A_t - \mathbb{E}[A_t]) \xrightarrow{d} N(0,1),$$

*where we define*

$$\sigma^2 \overset{\text{def.}}{=} \lim_{\ell \to \infty} \ell^{-1} \sum_{s=1}^{\ell} \left( \text{Var}_P(A_{t+s}) + 2 \sum_{i=1}^{m} \text{Cov}_P(A_{t+s}, A_{t+s-i}) \right).$$

Lemma C.6 says that the oracle estimator approaches the estimand at $\sqrt{T}$ rates and is asymptotically normal. The proof of Lemma C.6 applies Theorem C.4. To apply the CLT, we just need to verify that we can apply one of the two central limit theorems above: Theorem C.4 for Assumption 2.1(a) and Theorem C.5 for (b).

**Lemma C.6** (Corollary to Theorem C.4). *For an estimator $\psi$ satisfying Assumption 2.1 and Assumption C.3, it holds*

$$\sqrt{T} \sigma^{-1} (\psi(W_{1:T}; \eta^*) - \psi^*) \xrightarrow{d} N(0,1).$$

*Proof of Lemma C.6.* From Assumption 2.1, under part (a), we have that $\{W_t\}$ is a geometrically ergodic Markov chain. Also, $\varphi(W_t; \eta^*)$ is bounded in $L^{2+\delta}(P)$ by Assumption C.3, Equation (C.2). So $\psi(W_{1:T}; \eta^*) - \psi^*$ is bounded in probability by the triangle inequality:

$$\|\psi(W_{1:T}; \eta^*) - \psi^*\|_{L^{2+\delta}(P)} \leq T^{-1} \sum_{t=1}^{T} \|\varphi(W_t; \eta^*)\|_{L^{2+\delta}(P)} + |\psi^*| < \infty$$

Thus, the result under Assumption 2.1(a) follows directly from the application of Theorem C.4.

For Assumption 2.1(b), since $\{W_t\}_{t=1}^{\infty}$ is $m$-dependent, so is $\{\psi(W_t; \hat{\eta})\}_{t=1}^{\infty}$. By Assumption C.3, Equation (C.2), we have that $\varphi(W_t; \eta^*)$ is bounded in $L^3$. Thus, we can apply Theorem C.5.

$\square$

Lemma C.7 says that the deviation from its mean of the difference between the oracle estimator and our estimator approaches zero at faster-than-$\sqrt{T}$ rates. Our proof pattern is similar to that of Chernozhukov et al. (2018): our strategy is to prove that the variance of the expression goes to zero at faster-than-$T$ rates, so that we can apply Chebyshev's inequality, which implies that the expression itself goes to zero at faster-than-$\sqrt{T}$ rates. The core difficulty of proving the lemma, compared to the analogous result in Chernozhukov et al. (2018), is that we must account for the covariance between observations at different times. Chernozhukov et al. (2018) assumes independence between observations and thus does not have to account for such covariances. We handle the covariances separately for Assumption 2.1(a) and for Assumption 2.1(b). Under Assumption 2.1(a), we can invoke a theorem saying that geometrically ergodic Markov chains satisfying detailed balance have correlations between observations that decrease to zero at an exponential rate. This implies that the sum of correlations is bounded by a constant, so they do not dominate. Under Assumption 2.1(b), there are only finitely many non-zero correlations, so their sum is also bounded by a constant.

**Lemma C.7.** *Under Assumption 2.1, for an estimator $\psi$ and nuisance parameter estimators $\hat{\eta}$ satisfying Assumption C.2 and Assumption C.3, with probability $1 - \gamma$, it holds*

$$\left| \psi(W_{1:T}; \hat{\eta}) - \psi(W_{1:T}; \eta^*) - \mathbb{E}_P[\psi(W_{1:T}; \hat{\eta}) - \psi(W_{1:T}; \eta^*) \mid W^{\text{aux}\cdot}] \right| = o_P(T^{-1/2})$$

*Proof of Lemma C.7.* First, we will introduce new notation. For all $t \in [T]$, let

$$\hat{\varphi}_t \overset{\text{def.}}{=} \varphi(W_t; \hat{\eta}),$$
$$\varphi_t^* \overset{\text{def.}}{=} \varphi(W_t; \eta^*).$$

Let $\mathcal{E}_T$ be the event that $\hat{\eta} \in S_T$ (which happens with probability $1 - \gamma$ by Assumption C.3). Also, $\hat{\eta}$ is assumed to be a deterministic function of $W^{\mathrm{aux}\cdot}$. On the event $\mathcal{E}_T$,

$$\mathbb{E}_P \left[ (\psi(W_{1:T}; \hat{\eta}) - \psi(W_{1:T}; \eta^*) - \mathbb{E}_P[\psi(W_{1:T}; \hat{\eta}) - \psi(W_{1:T}; \eta^*) \mid W^{\mathrm{aux}\cdot}])^2 \mid W^{\mathrm{aux}\cdot} \right]$$

$$= \mathbb{E}_P \left[ \left( T^{-1} \sum_{t=1}^{T} \hat{\varphi}_t - \varphi_t^* - \mathbb{E}_P[\hat{\varphi}_t - \varphi_t^* \mid W^{\mathrm{aux}\cdot}] \right)^2 \mid W^{\mathrm{aux}\cdot} \right] \qquad \text{(Definition of } \psi.)$$

$$= T^{-2} \sum_{t=1}^{T} \mathbb{E}_P[(\hat{\varphi}_t - \varphi_t^* - \mathbb{E}_P[\hat{\varphi}_t - \varphi_t^* \mid W^{\mathrm{aux}\cdot}])^2 \mid W^{\mathrm{aux}\cdot}]$$

$$+ 2 \sum_{\ell=1}^{t-1} \mathbb{E}_P \big[ (\hat{\varphi}_t - \varphi_t^* - \mathbb{E}_P[\hat{\varphi}_t - \varphi_t^* \mid W^{\mathrm{aux}\cdot}]) \qquad\qquad \text{(C.8)}$$

$$\cdot (\hat{\varphi}_{t-\ell} - \varphi_{t-\ell}^* - \mathbb{E}_P[\hat{\varphi}_{t-\ell} - \varphi_{t-\ell}^* \mid W^{\mathrm{aux}\cdot}]) \mid W^{\mathrm{aux}\cdot} \big] \qquad \text{(Rearranging.)}$$

$$\leq \sup_{\eta \in S_T} T^{-2} \sum_{t=1}^{T} \mathbb{E}_P[(\varphi(W_t; \eta) - \varphi_t^* - \mathbb{E}_P[\varphi(W_t; \eta) - \varphi_t^* \mid W^{\mathrm{aux}\cdot}])^2 \mid W^{\mathrm{aux}\cdot}]$$

$$+ 2 \sum_{\ell=1}^{t-1} \mathbb{E}_P \big[ (\varphi(W_t; \eta) - \varphi_t^* - \mathbb{E}_P[\varphi(W_t; \eta) - \varphi_t^* \mid W^{\mathrm{aux}\cdot}])$$

$$\cdot (\varphi(W_{t-\ell}; \eta) - \varphi_{t-\ell}^* - \mathbb{E}_P[\varphi(W_{t-\ell}; \eta) - \varphi_{t-\ell}^* \mid W^{\mathrm{aux}\cdot}]) \mid W^{\mathrm{aux}\cdot} \big] \qquad (\hat{\eta} \in S_T \text{ on } \mathcal{E}_T)$$

$$= \sup_{\eta \in S_T} T^{-2} \sum_{t=1}^{T} \mathbb{E}_P[(\varphi(W_t; \eta) - \varphi_t^* - \mathbb{E}_P[\varphi(W_t; \eta) - \varphi_t^*])^2]$$

$$+ 2 \sum_{\ell=1}^{t-1} \mathbb{E}_P \big[ (\varphi(W_t; \eta) - \varphi_t^* - \mathbb{E}_P[\varphi(W_t; \eta) - \varphi_t^*]) \qquad\qquad \text{(C.9)}$$

$$\cdot (\varphi(W_{t-\ell}; \eta) - \varphi_{t-\ell}^* - \mathbb{E}_P[\varphi(W_{t-\ell}; \eta) - \varphi_{t-\ell}^*]) \big]$$

where the last equality comes from the independence of $W^{\mathrm{aux}\cdot}$ from $W_{1:T}$. Now, for all $t \in \mathbb{N}$, let $\mathcal{F}_t^\infty$ be the $\sigma$-algebra generated by $W_{t:\infty}$ and let $\mathcal{F}_1^t$ be the $\sigma$-algebra generated by $W_{1:t}$. Recall the definition of $\rho$-mixing (see, e.g., (Bradley, 2007)):

$$\rho(t) \overset{\mathrm{def.}}{=} \sup_{f \in L^2(\mathcal{F}_i^\infty), g \in L^2(\mathcal{F}_1^{i-t})} |\mathrm{Corr}(f, g)|,$$

where the correlation, covariance and variance for functions has the usual definition:

$$\mathrm{Corr}(f, g) = \mathrm{Cov}(f, g) / (\mathrm{Var}(f)\mathrm{Var}(g))^{1/2},$$
$$\mathrm{Cov}(f, g) = \mathbb{E}_P \big[ (f(W_{i:\infty}) - \mathbb{E}_P[f(W_{i:\infty})])(g(W_{1:(i-t)})) - \mathbb{E}_P[g(W_{1:(i-t)})]) \big],$$

and $\mathrm{Var}(f) = \mathrm{Cov}(f, f)$. Next, we state a lemma that will allow us to bound the covariance between terms in the above sum under Assumption 2.1(a).

**Lemma C.8** (Theorem 2, (Jones, 2005)). *If a Markov chain $A_{1:\infty}$ is geometrically ergodic and satisfies detailed balance, then it is $\rho$-mixing with $\rho(T) \leq O(e^{-cT})$ for some $c > 0$.*

On the other hand, under Assumption 2.1(b), $W_{1:\infty}$ is $\rho$ mixing with $\rho(\ell) = 0$ for all $\ell > m$. Note that these facts imply $W_{1:\infty}$ is $\rho$-mixing with $\rho(t) = O(e^{-cT})$ for some $c > 0$. Also, for all $t \in [T]$, $\eta \in S_T$ and $\ell < t$, it holds $\varphi(W_t; \eta) - \varphi_t^* - \mathbb{E}_P[\varphi(W_t; \eta) - \varphi_t^*] \in L^2(\mathcal{F}_t^\infty)$ and $\varphi(W_{t-\ell}; \eta) - \varphi_{t-\ell}^* - \mathbb{E}_P[\varphi(W_{t-\ell}; \eta) - \varphi_{t-\ell}^*] \in L^2(\mathcal{F}_1^{t-\ell})$ (where expectations are taken with respect to $P$). To see this, note that for all $t \in [T]$ and $\eta \in S_T$, it holds $\|\varphi(W_t; \eta)\|_{L^2(P)} < \infty$ by Assumption C.3, Equation (C.2). Also, since the expressions just depend on $W_t$ and $W_{t-\ell}$, respectively, this implies $\|\varphi(W_t; \eta)\|_{L^2(P)} = \|\varphi(W_t; \eta)\|_{L^2(\mathcal{F}_t^\infty)}$ and similarly $\|\varphi(W_{t-\ell}; \eta)\|_{L^2(P)} = \|\varphi(W_{t-\ell}; \eta)\|_{L^2(\mathcal{F}_1^{t-\ell})}$. Also, it is assumed $\eta^* \in S_T$, so by the same reasoning $\|\varphi_t^*\|_{L^2(\mathcal{F}_t^\infty)} < \infty$ and $\|\varphi_{t-\ell}^*\|_{L^2(\mathcal{F}_1^{t-\ell})} < \infty$.

Thus,

$$\text{Corr}\left(\varphi(W_t; \eta) - \varphi_t^* - \mathbb{E}_P[\varphi(W_t; \eta) - \varphi_t^*], \varphi(W_{t-\ell}; \eta) - \varphi_{t-\ell}^* - \mathbb{E}_P[\varphi(W_{t-\ell}; \eta) - \varphi_{t-\ell}^*]\right) \leq \rho(\ell).$$

This implies we can rewrite Equation (C.9) as

$$\sup_{\eta \in S_T} T^{-2} \sum_{t=1}^{T} \mathbb{E}_P[(\varphi(W_t; \eta) - \varphi_t^* - \mathbb{E}_P[\varphi(W_t; \eta) - \varphi_t^*])^2]$$

$$+ 2 \sum_{\ell=1}^{t-1} \mathbb{E}_P\big[ \left(\varphi(W_t; \eta) - \varphi_t^* - \mathbb{E}_P[\varphi(W_t; \eta) - \varphi_t^*]\right)$$

$$\cdot \left(\varphi(W_{t-\ell}; \eta) - \varphi_{t-\ell}^* - \mathbb{E}_P[\varphi(W_{t-\ell}; \eta) - \varphi_{t-\ell}^*]\right) \big]$$

$$\leq \sup_{\eta \in S_T} T^{-2} \sum_{t=1}^{T} \left\| \varphi(W_t; \eta) - \varphi_t^* - \mathbb{E}_P[\varphi(W_t; \eta) - \varphi_t^*] \right\|_{L^2(P)}^2$$

$$+ 2 \sum_{\ell=1}^{t-1} \rho(\ell) \left\| \varphi(W_t; \eta) - \varphi_t^* - \mathbb{E}_P[\varphi(W_t; \eta) - \varphi_t^*] \right\|_{L^2(P)}$$

$$\cdot \left\| \varphi(W_{t-\ell}; \eta) - \varphi_t^* - \mathbb{E}_P[\varphi(W_{t-\ell}; \eta) - \varphi_{t-\ell}^*] \right\|_{L^2(P)}$$

$$\leq \sup_{\eta \in S_T} T^{-2} \sum_{t=1}^{T} \left\| \varphi(W_t; \eta) - \varphi_t^* \right\|_{L^2(P)}^2$$

$$+ 2 \sum_{\ell=1}^{t-1} \rho(\ell) \left\| \varphi(W_t; \eta) - \varphi_t^* \right\|_{L^2(P)} \left\| \varphi(W_{t-\ell}; \eta) - \varphi_{t-\ell}^* \right\|_{L^2(P)}$$

Now, notice

$$T^{-1} \sum_{t=1}^{T} \left\| \varphi(W_t; \eta) - \varphi_t^* \right\|_{L^2(P)}^2 = o(1)$$

by Assumption C.3, Equation (C.3), so

$$T^{-2} \sum_{t=1}^{T} \left\| \varphi(W_t; \eta) - \varphi_t^* \right\|_{L^2(P)}^2 = o(T^{-1}).$$

Next, since $\rho(\ell) = O(e^{-c\ell})$ for some $c > 0$ (by Lemma C.8) and

$$T^{-1} \sum_{t=1}^{T} \left\| \varphi(W_t; \eta) - \varphi_t^* \right\|_{L^2(P)} = o(1)$$

for all $t \in [T]$, it holds

$$T^{-2} \sum_{t=1}^{T} \sum_{\ell=1}^{t-1} \rho(\ell) \left\| \varphi(W_t; \eta) - \varphi_t^* \right\|_{L^2(P)} \left\| \varphi(W_{t-\ell}; \eta) - \varphi_{t-\ell}^* \right\|_{L^2(P)}$$

$$= T^{-2} \sum_{\ell=1}^{T} \rho(\ell) \sum_{t=\ell+1}^{T} \left\| \varphi(W_{t-\ell}; \eta) - \varphi_{t-\ell}^* \right\|_{L^2(P)} \left\| \varphi(W_t; \eta) - \varphi_t^* \right\|_{L^2(P)} \qquad \text{(Exchanging sums.)}$$

$$= T^{-2} \sum_{\ell=1}^{T} \rho(\ell) \left( \sum_{t=\ell+1}^{T} \left\| \varphi(W_{t-\ell}; \eta) - \varphi_{t-\ell}^* \right\|_{L^2(P)} \right)^{1/2} \left( \sum_{t=\ell+1}^{T} \left\| \varphi(W_t; \eta) - \varphi_t^* \right\|_{L^2(P)} \right)^{1/2}$$

$$\text{(Cauchy-Schwarz inequality.)}$$

$$= T^{-1} \sum_{\ell=1}^{T} \rho(\ell) \left( T^{-1} \sum_{t=\ell+1}^{T} \left\| \varphi(W_{t-\ell}; \eta) - \varphi_{t-\ell}^* \right\|_{L^2(P)} \right)^{1/2} \left( T^{-1} \sum_{t=\ell+1}^{T} \left\| \varphi(W_t; \eta) - \varphi_t^* \right\|_{L^2(P)} \right)^{1/2}$$

$$\text{(Distributing } T^{-1}.\text{)}$$

$$= T^{-1} \sum_{\ell=1}^{T} \rho(\ell) o(1) \qquad\qquad (T^{-1} \sum_{t=\ell+1}^{T} \|\varphi(W_t; \eta) - \varphi_t^*\|_{L^2(P)} = o(1).)$$

$$= T^{-1} \sum_{\ell=1}^{T} O(e^{-c\ell}) o(1) \qquad\qquad (\rho(\ell) = O(e^{-c\ell}) \text{ for some } c > 0.)$$

$$\leq o(T^{-1}).$$

where the last line comes from the fact that $\sum_{t=1}^{T} e^{-\theta t} \leq 1/(1 - e^{-\theta})$. Thus, applying Chebyshev's inequality, we have

$$(\psi(W_{1:T}; \hat\eta) - \psi(W_{1:T}; \eta^*) - \mathbb{E}_P[\psi(W_{1:T}; \hat\eta) - \psi(W_{1:T}; \eta^*) \mid W^{\text{aux}\cdot}]) \mid W^{\text{aux}\cdot} = o_P(T^{-1/2}).$$

Finally, the fact that conditional convergence implies unconditional convergence (Lemma 6.1 of (Chernozhukov et al., 2018)) implies the desired result. $\qquad\square$

Lemma C.9 says that the absolute value of the expectation of the difference between the estimator and oracle goes to zero at faster-than-$\sqrt{T}$ rates. The proof is nearly identical to the analogous argument in Chernozhukov et al. (2018): we apply Taylor's theorem by differentiating on the path between $\hat\eta$ and $\eta^*$ to show that small deviations of nuisances $\eta$ around $\eta^*$ do not dramatically affect the deviation of the estimator from the oracle estimator. The first-order term is zero by the assumption of Neyman orthogonality. The second-order term is small by assumption (Equation (C.4)).

**Lemma C.9.** *For an estimator $\psi$ and nuisance parameter estimators $\hat\eta$ satisfying Assumption C.2 and Assumption C.3, it holds*

$$\left| \mathbb{E}_P\left[\psi(W_{1:T}; \hat\eta) - \psi(W_{1:T}; \eta^*) \mid W^{\text{aux}\cdot}\right] \right| = o_P(T^{-1/2}).$$

*Proof of Lemma C.9.* We use Neyman orthogonality. Define

$$f(r) \stackrel{\text{def.}}{=} \mathbb{E}_P\left[\psi(W_{1:T}; \eta^* + r(\hat\eta - \eta^*)) \mid W^{\text{aux}\cdot}\right] - \mathbb{E}_P\left[\psi(W_{1:T}; \eta^*)\right], \qquad r \in [0, 1]$$

Then, notice

$$f(1) = \mathbb{E}_P\left[\psi(W_{1:T}; \hat\eta) \mid W^{\text{aux}\cdot}\right] - \mathbb{E}_P\left[\psi(W_{1:T}; \eta^*)\right]$$

is the quantity we want to bound in probability. By Taylor's Theorem

$$f(1) = f(0) + \frac{\partial}{\partial r} f(0) + \frac{\partial^2}{\partial r^2} f(\tilde{r})/2$$

for some $\tilde{r} \in (0, 1)$. Notice that $f(0) = 0$ since $\psi(W_{1:T}; \eta^*)$ is independent of $W^{\text{aux}\cdot}$. Also, notice that $\partial f(0)/\partial r = 0$ by Assumption C.2(b). Finally, on the event $\mathcal{E}_T$,

$$\left| \frac{\partial^2}{\partial r^2} f(\tilde{r}) \right| \leq \sup_{r \in (0,1)} \left| \frac{\partial^2}{\partial r^2} f(r) \right| = o_P(T^{-1/2})$$

where the convergence comes from Equation (C.4). $\qquad\square$

Lemma C.10 states the consistency of our variance estimator $\hat\sigma^2$. We prove consistency for the estimators under Assumption 2.1(a) and Assumption 2.1(b) separately. For Assumption 2.1(a), we can directly apply a consistent variance estimation result from prior work. For Assumption 2.1(b), we prove consistency of the plugin variance estimator directly. We handle the variance terms and covariance terms in $\psi$ separately. For the variance terms, we follow the argument in Chernozhukov et al. (2018). For the covariance terms, we follow a similar pattern to the argument for the variance terms: show that the average of the covariances converge in probability to covariance estimates computed using oracle nuisances, and then showing that the covariance estimates computed using oracle nuisances converge to their expectation.

**Lemma C.10.** *For $\hat\sigma_T^2$ as defined in Equation (C.1) and $\sigma^2 = \lim_{t \to \infty} \text{Var}_P(\psi(W_{1:T}, \hat\eta))$ and under Assumptions 2.1, C.2 and C.3, it holds $\hat\sigma_T^2 \xrightarrow{p} \sigma^2$.*

*Proof.* We split the analysis into two cases. The first case handles when Assumption 2.1(a) holds and the second handles when Assumption 2.1(b) holds.

**Case 1: Under Assumption 2.1(a).** We will apply the following result adapted from (Jones et al., 2006). For simplicity, we will use $T_1, T_2$ as defined in the statement of the result.

**Lemma C.11** (Proposition 3, (Jones et al., 2006)). *Suppose a Markov chain $\{A_t\}_{t=1}^\infty$ is geometrically ergodic with stationary distribution $\pi$. Also suppose that there exists a constant $\delta$ such that a measurable function $f$ satisfies $\|f\|_{L^{2+\delta}(\pi)} < C$. Define*

$$\sigma^2 \overset{\text{def.}}{=} \operatorname{Var}_\pi(f(A_1)) + 2\sum_{t=1}^\infty \operatorname{Cov}_\pi(f(A_1), f(A_t)).$$

*and*

$$\hat{\sigma}^2 \overset{\text{def.}}{=} \frac{1}{T_2(T_1-1)} \sum_{t=0}^{T_1-1} \left( \sum_{s=tT_2+1}^{(t+1)T_2} f(A_t) - T^{-1} \sum_{i=1}^T f(A_t) \right)^2.$$

*Then it holds $\hat{\sigma}^2 \to \sigma^2$.*

Notice that by Assumption C.3, Equation (C.2) and the fact that $K_\infty \in \mathcal{P}$, we have assumed $\|\varphi(W_t; \eta)\|_{L^{4+\delta}(K^\infty)} < C$. Thus, we can apply Lemma C.11 and the result under Assumption 2.1(a) follows.

**Case 2: Under Assumption 2.1(b)** Define

$$\sigma_T^2 \overset{\text{def.}}{=} \sum_{t=1}^{T-m} \operatorname{Var}(\varphi(W_{m+t}; \eta^*)) + 2\sum_{i=1}^m \operatorname{Cov}(\varphi(W_{m+t}; \eta^*), \varphi(W_{m+t-i}; \eta^*))$$

We will prove

$$\hat{\sigma}_T^2 - \sigma_T^2 \overset{p}{\to} 0$$

which will imply the result, since $\sigma_T^2 \to \sigma^2$ trivially. Using the notation from Lemma C.7, notice

$$\hat{\sigma}_T^2 - \sigma_T^2 = T^{-1} \sum_{t=1}^T (\hat{\varphi}_t - \psi(W_{1:T}; \hat{\eta}))^2 - \mathbb{E}\left[(\varphi_t^* - \psi^*)^2\right] \tag{C.10}$$

$$+ 2T^{-1} \sum_{t=1}^T \sum_{i=1}^{\min\{t-1,m\}} (\hat{\varphi}_t - \psi(W_{1:T}; \hat{\eta}))(\hat{\varphi}_{t-i} - \psi(W_{1:T}; \hat{\eta})) - \mathbb{E}\left[(\varphi_t^* - \psi^*)(\varphi_{t-i}^* - \psi^*)\right] \tag{C.11}$$

by definition. We will bound Equation (C.10) and Equation (C.11) separately. First, for Equation (C.10),

$$T^{-1} \sum_{t=1}^T (\hat{\varphi}_t - \psi(W_{1:T}; \hat{\eta}))^2 - \mathbb{E}\left[(\varphi_t^* - \psi^*)^2\right]$$

$$= T^{-1} \sum_{t=1}^T (\hat{\varphi}_t - \psi(W_{1:T}; \hat{\eta}))^2 - (\varphi_t^* - \psi^*)^2 \tag{C.12}$$

$$+ (\varphi_t^* - \psi^*)^2 - \mathbb{E}\left[(\varphi_t^* - \psi^*)^2\right] \tag{C.13}$$

by adding and subtracting terms. Next, to bound Equation (C.12) in probability, notice

$$\left( T^{-1} \sum_{t=1}^T (\hat{\varphi}_t - \psi(W_{1:T}; \hat{\eta}))^2 - (\varphi_t^* - \psi^*)^2 \right)^2$$

$$= T^{-2} \left( \sum_{t=1}^T (\hat{\varphi}_t - \psi(W_{1:T}; \hat{\eta}) - \varphi_t^* + \psi^*)(\hat{\varphi}_t - \psi(W_{1:T}; \hat{\eta}) + \varphi_t^* - \psi^*) \right)^2$$

$$\leq T^{-1}\sum_{t=1}^{T}\left(\hat{\varphi}_t - \psi(W_{1:T};\hat{\eta}) - \varphi_t^* + \psi^*\right)^2 \tag{C.14}$$

$$\cdot T^{-1}\sum_{t=1}^{T}\left(\hat{\varphi}_t - \psi(W_{1:T};\hat{\eta}) + \varphi_t^* - \psi^*\right)^2 \tag{C.15}$$

The first (in)equality comes from rearranging; the second comes from the Cauchy-Schwarz inequality. Now, we can bound Equation (C.14) as

$$T^{-1}\sum_{t=1}^{T}\left(\hat{\varphi}_t - \psi(W_{1:T};\hat{\eta}) - \varphi_t^* + \psi^*\right)^2$$

$$\leq 2T^{-1}\left(\sum_{t=1}^{T}(\hat{\varphi}_t - \varphi_t^*)^2\right) + 2\left(\psi(W_{1:T};\hat{\eta}) - \psi^*\right)^2$$

$$\leq 2T^{-1}\left(\sum_{t=1}^{T}(\hat{\varphi}_t - \varphi_t^*)^2\right) \tag{C.16}$$

$$+ 4\left(\psi(W_{1:T};\hat{\eta}) - \mathbb{E}_P\left[\psi(W_{1:T};\hat{\eta}) \mid W^{\text{aux.}}\right]\right)^2 \tag{C.17}$$

$$+ 4\mathbb{E}_P\left[\psi(W_{1:T};\hat{\eta}) - \psi^* \mid W^{\text{aux.}}\right]^2 \tag{C.18}$$

where the first inequality comes from the fact that $(a+b)^2 \leq 2(a^2+b^2)$ for all $a,b$; the second comes from adding and subtracting the conditional expectation and then again applying $(a+b)^2 \leq 2(a^2+b^2)$. Equation (C.16) is $o_P(1)$ by Markov's inequality and Equation (C.3) since

$$T^{-1}\left(\sum_{t=1}^{T}\mathbb{E}_P\left[(\hat{\varphi}_t - \varphi_t^*)^2\right]\right) = T^{-1}\sum_{t=1}^{T}\|\hat{\varphi}_t - \varphi_t^*\|_{L^2(P)}^2$$

$$= T^{-1}\sum_{t=1}^{T}o(1).$$

Equation (C.17) is $o_P(1)$ by the following sequence.

$$\left(\psi(W_{1:T};\hat{\eta}) - \mathbb{E}_P\left[\psi(W_{1:T};\hat{\eta}) \mid W^{\text{aux.}}\right]\right)^2$$

$$= T^{-2}\left(\sum_{t=1}^{T}\hat{\varphi}_t - \mathbb{E}_P\left[\hat{\varphi}_t\right]\right)^2 \qquad \text{(Definition of } \psi.\text{)}$$

$$= T^{-2}\sum_{t=1}^{T}(\hat{\varphi}_t - \mathbb{E}_P\left[\hat{\varphi}_t\right])^2 + 2\sum_{i=1}^{m}(\hat{\varphi}_t - \mathbb{E}_P\left[\hat{\varphi}_t\right])(\hat{\varphi}_{t-i} - \mathbb{E}_P\left[\hat{\varphi}_{t-i}\right]) \qquad \text{(Rearranging.)}$$

$$\leq T^{-2}\sum_{t=1}^{T}(\hat{\varphi}_t - \mathbb{E}_P\left[\hat{\varphi}_t\right])^2 + 2\sqrt{m}(\hat{\varphi}_t - \mathbb{E}_P\left[\hat{\varphi}_t\right])\left(\sum_{i=1}^{m}(\hat{\varphi}_{t-i} - \mathbb{E}_P\left[\hat{\varphi}_{t-i}\right])^2\right)^{1/2} \qquad \text{(Cauchy-Schwarz inequality.)}$$

$$= T^{-2}\sum_{t=1}^{T}O_P(1) + 2\sqrt{m}O_P(1)\left(\sum_{i=1}^{m}O_P(1)\right)^{1/2}$$

The last line comes by Markov's inequality and the fact that $\mathbb{E}_P\left[(\hat{\varphi}_t - \mathbb{E}_P\left[\hat{\varphi}_t\right])^2\right] = O(1)$ for all $t$ by Equation (C.2) and the fact that $m$ is a constant independent of $T$. Equation (C.18) is $o(1)$ by the following sequence.

$$\mathbb{E}_P\left[\psi(W_{1:T};\hat{\eta}) - \psi^* \mid W^{\text{aux.}}\right]^2 = \mathbb{E}_P\left[\psi(W_{1:T};\hat{\eta}) - \psi(W_{1:T};\eta^*) \mid W^{\text{aux.}}\right]^2$$

$$= T^{-2}\left(\sum_{t=1}^{T}\mathbb{E}_P\left[\hat{\varphi}_t - \varphi_t^*\right]\right)^2$$

$$= T^{-2} \left( \sum_{t=1}^{T} o(1) \right)^2$$

$$= o(1)$$

The first line comes by the fact that $\mathbb{E}_P \left[ \psi(W_{1:T}; \eta^*) \mid W^{\mathrm{aux}\cdot} \right] = \psi^*$, the second is by definition of $\psi$, and the last is Equation (C.3). Thus, Equation (C.14) is $o_P(1)$. We can bound Equation (C.15) in probability as

$$T^{-1} \sum_{t=1}^{T} \left( \hat{\varphi}_t - \psi(W_{1:T}; \hat{\eta}) + \varphi_t^* - \psi^* \right)^2$$

$$\leq 2T^{-1} \left( \sum_{t=1}^{T} (\hat{\varphi}_t + \varphi_t^*)^2 \right) + 2(\psi(W_{1:T}; \hat{\eta}) + \psi^*)^2$$

$$\leq 4T^{-1} \left( \sum_{t=1}^{T} \hat{\varphi}_t^2 + \varphi_t^{*2} \right) + 4T^{-2} \left( \sum_{t=1}^{T} \hat{\varphi}_t \right)^2 + 4\psi^{*2}$$

$$\leq \sup_{\eta \in S_T} 4T^{-1} \left( \sum_{t=1}^{T} \varphi(W_t; \eta)^2 + \varphi_t^{*2} \right) + 4T^{-2} \left( \sum_{t=1}^{T} \varphi(W_t; \eta) \right)^2 + 4\psi^{*2}$$

$$= O_P(1) \tag{C.19}$$

where in each inequality, we apply the fact that $(a + b)^2 \leq 2a^2 + 2b^2$ and in the last line we apply Markov's inequality with Equation (C.2): Namely, from Equation (C.2) there exists a constant $C$ such that, for all $s, t \in [T]$,

$$\sup_{\eta \in S_T} \mathbb{E} \left[ \varphi(W_t; \eta)^2 \right] = \| \varphi(W_t; \eta) \|_{L^2(P)}^2 \leq C,$$

$$\mathbb{E} \left[ \varphi_t^{*2} \right] = \| \varphi_t^* \|_{L^2(P)}^2 \leq C, \text{ and}$$

$$\sup_{\eta \in S_T} \mathbb{E} \left[ \varphi(W_t; \eta) \varphi(W_s; \eta) \right] \leq \| \varphi(W_t; \eta) \|_{L^2(P)} \| \varphi(W_s; \eta) \|_{L^2(P)} \leq C.$$

(The last line applies the Cauchy-Schwarz inequality.) The fact that Equations (C.14) and (C.15) are $o_P(1)$ and $O_P(1)$ respectively imply that the expression in Equation (C.12) is $o_P(1)$. Next, we bound Equation (C.13) in probability. Notice

$$\mathbb{E} \left[ \left( T^{-1} \sum_{t=1}^{T} (\varphi_t^* - \psi^*)^2 - \mathbb{E} \left[ (\varphi_t^* - \psi^*)^2 \right] \right)^2 \right]$$

$$= T^{-2} \sum_{t=1}^{T} \mathbb{E} \left[ \left( (\varphi_t^* - \psi^*)^2 - \mathbb{E} \left[ (\varphi_t^* - \psi^*)^2 \right] \right)^2 \right]$$

$$+ \sum_{i=1}^{m} \mathbb{E} \left[ \left( (\varphi_t^* - \psi^*)^2 - \mathbb{E} \left[ (\varphi_t^* - \psi^*)^2 \right] \right) \left( (\varphi_{t-i}^* - \psi^*)^2 - \mathbb{E} \left[ (\varphi_{t-i}^* - \psi^*)^2 \right] \right) \right]$$

$$\leq T^{-2} \sum_{t=1}^{T} \| \varphi_t^* - \psi^* \|_{L^4(P)}^4$$

$$+ \sqrt{m} \| \varphi_t^* - \psi^* \|_{L^2(P)}^2 \left( \sum_{i=1}^{m} \| \varphi_{t-i}^* - \psi^* \|_{L^2(P)}^2 \right)^{1/2}$$

$$\leq T^{-2} \sum_{t=1}^{T} \| \varphi_t^* - \psi^* \|_{L^4(P)}^4$$

$$+ \sqrt{m} \| \varphi_t^* - \psi^* \|_{L^2(P)}^2 \left( \sum_{i=1}^{m} \| \varphi_{t-i}^* - \psi^* \|_{L^2(P)}^2 \right)^{1/2}$$

$$= o_P(1)$$

where in the last line we apply the triangle inequality $\|\varphi_t^* - \psi^*\|_{L^4(P)} \leq \|\varphi_t^*\|_{L^4(P)} + |\psi^*|$ and Equation (C.2). Thus, the fact that Equation (C.12) is $o_P(1)$ and that Equation (C.13) is $o_P(1)$ imply that Equation (C.10) is $o_P(1)$. To bound Equation (C.11),

$$T^{-1} \sum_{t=1}^{T} \sum_{i=1}^{m} \left(\hat{\varphi}_t - \psi(W_{1:T}; \hat{\eta})\right) \left(\varphi_{t-i}^* - \psi(W_{1:T}; \hat{\eta})\right) - \mathbb{E}\left[(\varphi_t^* - \psi^*)(\varphi_{t-i}^* - \psi^*)\right]$$

$$= T^{-1} \sum_{t=1}^{T} \sum_{i=1}^{m} \left(\hat{\varphi}_t - \psi(W_{1:T}; \hat{\eta})\right) \left(\hat{\varphi}_{t-i} - \psi(W_{1:T}; \hat{\eta})\right) - \left(\varphi_t^* - \psi^*\right) \left(\varphi_{t-i}^* - \psi^*\right) \tag{C.20}$$

$$+ \left(\varphi_t^* - \psi^*\right) \left(\varphi_{t-i}^* - \psi^*\right) - \mathbb{E}\left[(\varphi_t^* - \psi^*)(\varphi_{t-i}^* - \psi^*)\right] \tag{C.21}$$

To bound Equation (C.20),

$$T^{-1} \sum_{t=1}^{T} \sum_{i=1}^{m} \left(\hat{\varphi}_t - \psi(W_{1:T}; \hat{\eta})\right) \left(\hat{\varphi}_{t-i} - \psi(W_{1:T}; \hat{\eta})\right) - \left(\varphi_t^* - \psi^*\right) \left(\varphi_{t-i}^* - \psi^*\right)$$

$$= T^{-1} \sum_{t=1}^{T} \sum_{i=1}^{m} \left(\hat{\varphi}_t - \psi(W_{1:T}; \hat{\eta}) - \varphi_t^* + \psi^*\right) \left(\hat{\varphi}_{t-i} - \psi(W_{1:T}; \hat{\eta})\right)$$

$$+ \left(\varphi_t^* - \psi^*\right) \left(\hat{\varphi}_{t-i} - \psi(W_{1:T}; \hat{\eta}) - \varphi_{t-i}^* + \psi^*\right)$$

$$= T^{-1} \sum_{t=1}^{T} \left(\hat{\varphi}_t - \psi(W_{1:T}; \hat{\eta}) - \varphi_t^* + \psi^*\right) \sum_{i=1}^{m} \left(\hat{\varphi}_{t-i} - \psi(W_{1:T}; \hat{\eta})\right)$$

$$+ T^{-1} \sum_{t=1}^{T} \left(\varphi_t^* - \psi^*\right) \sum_{i=1}^{m} \left(\hat{\varphi}_{t-i} - \psi(W_{1:T}; \hat{\eta}) - \varphi_{t-i}^* + \psi^*\right)$$

$$= T^{-1} \sum_{t=1}^{T} o_P(1) \cdot \sum_{i=1}^{m} \left(\hat{\varphi}_{t-i} - \psi(W_{1:T}; \hat{\eta})\right)$$

$$+ T^{-1} \sum_{t=1}^{T} \left(\varphi_t^* - \psi^*\right) \sum_{i=1}^{m} o_P(1)$$

The first equality comes from the fact that for all $a, b, a', b'$, it holds $aa' - bb' = (a-b)a' + (a'-b')b$; the second comes from rearranging; the third comes from Equation (C.3). Now, we just need to argue that $\varphi_t^* - \psi^* = O_P(1)$ and $\hat{\varphi}_t - \psi(W_{1:T}; \hat{\eta}) = O_P(1)$ for all $t$ to show that that Equation (C.20) is $o_P(1)$.

$$\mathbb{E}\left[(\varphi_t^* - \psi^*)^2\right] \leq 2\|\varphi_t^*\|_{L^2(P)}^2 + 2\psi^{*2} \leq C,$$

$$\mathbb{E}\left[(\hat{\varphi}_t - \psi(W_{1:T}; \hat{\eta}))^2\right] \leq 2\|\hat{\varphi}_t\|_{L^2(P)}^2 + 2\|\psi(W_{1:T}; \hat{\eta}))\|_{L^2(P)}^2 \leq C$$

by Equation (C.2) so Chebyshev's inequality completes the argument. To bound Equation (C.21),

$$\mathbb{E}\left[\left(T^{-1} \sum_{t=1}^{T} \sum_{i=1}^{m} \left(\varphi_t^* - \psi^*\right) \left(\varphi_{t-i}^* - \psi^*\right) - \mathbb{E}\left[(\varphi_t^* - \psi^*)(\varphi_{t-i}^* - \psi^*)\right]\right)^2\right]$$

$$= T^{-2} \sum_{t=1}^{T} \sum_{i=1}^{m} \mathbb{E}_P\left[\left(\left(\varphi_t^* - \psi^*\right) \left(\varphi_{t-i}^* - \psi^*\right) - \mathbb{E}\left[(\varphi_t^* - \psi^*)(\varphi_{t-i}^* - \psi^*)\right]\right)^2\right]$$

$$+ 2 \sum_{j=1}^{m+i} \sum_{\ell=1}^{m+i} \mathbb{E}_P\left[\left(\left(\varphi_t^* - \psi^*\right) \left(\varphi_{t-i}^* - \psi^*\right) - \mathbb{E}\left[(\varphi_t^* - \psi^*)(\varphi_{t-i}^* - \psi^*)\right]\right)\right.$$

$$\left. \cdot \left(\left(\varphi_{t-j}^* - \psi^*\right) \left(\varphi_{t-\ell}^* - \psi^*\right) - \mathbb{E}\left[(\varphi_{t-j}^* - \psi^*)(\varphi_{t-\ell}^* - \psi^*)\right]\right)\right] \quad \text{(Rearranging; } m\text{-dependence.)}$$

$$\leq T^{-2} \sum_{t=1}^{T} \sum_{i=1}^{m} \mathbb{E}_P\left[\left(\left(\varphi_t^* - \psi^*\right) \left(\varphi_{t-i}^* - \psi^*\right)\right)^2\right]$$

$$+ 2 \sum_{j=1}^{m+i} \sum_{\ell=1}^{m+i} \left( \mathbb{E}_P \left[ \left( \left( \varphi_t^* - \psi^* \right) \left( \varphi_{t-i}^* - \psi^* \right) - \mathbb{E} \left[ (\varphi_t^* - \psi^*)(\varphi_{t-i}^* - \psi^*) \right] \right)^2 \right] \right.$$

$$\left. \cdot \mathbb{E}_P \left[ \left( \left( \varphi_{t-j}^* - \psi^* \right) \left( \varphi_{t-\ell}^* - \psi^* \right) - \mathbb{E} \left[ (\varphi_{t-j}^* - \psi^*)(\varphi_{t-\ell}^* - \psi^*) \right] \right)^2 \right] \right)^{1/2}$$

(For r.v. $X$, $\mathbb{E}(X - \mathbb{E}(X))^2 \leq \mathbb{E}(X^2)$; Cauchy-Schwarz inequality.)

$$\leq T^{-2} \sum_{t=1}^{T} \sum_{i=1}^{m} \left( \mathbb{E}_P \left[ (\varphi_t^* - \psi^*)^4 \right] \mathbb{E}_P \left[ (\varphi_{t-i}^* - \psi^*)^4 \right] \right)^{1/2}$$

$$+ 2 \sum_{j=1}^{m+i} \sum_{\ell=1}^{m+i} \left( \mathbb{E}_P \left[ \left( \left( \varphi_t^* - \psi^* \right) \left( \varphi_{t-i}^* - \psi^* \right) \right)^2 \right] \right.$$

$$\left. \cdot \mathbb{E}_P \left[ \left( \left( \varphi_{t-j}^* - \psi^* \right) \left( \varphi_{t-\ell}^* - \psi^* \right) \right)^2 \right] \right)^{1/2} \tag{—"—}$$

$$\leq T^{-2} \sum_{t=1}^{T} \sum_{i=1}^{m} \left( \mathbb{E}_P \left[ (\varphi_t^* - \psi^*)^4 \right] \mathbb{E}_P \left[ (\varphi_{t-i}^* - \psi^*)^4 \right] \right)^{1/2}$$

$$+ 2 \sum_{j=1}^{m+i} \sum_{\ell=1}^{m+i} \left( \mathbb{E}_P \left[ (\varphi_t^* - \psi^*)^4 \right] \mathbb{E}_P \left[ (\varphi_{t-i}^* - \psi^*)^4 \right] \right.$$

$$\left. \cdot \mathbb{E}_P \left[ (\varphi_{t-j}^* - \psi^*)^4 \right] \mathbb{E}_P \left[ (\varphi_{t-\ell}^* - \psi^*)^4 \right] \right)^{1/2} \tag{—"—}$$

$$\leq T^{-2} \sum_{t=1}^{T} \sum_{i=1}^{m} \left( \left( 8 \left\| \varphi_t^* \right\|_{L^4(P)}^4 + 8 \psi^{*4} \right) \left( \left\| 8 \varphi_{t-i}^* \right\|_{L^4(P)}^4 + 8 \psi^{*4} \right) \right)^{1/2}$$

$$+ 2 \sum_{j=1}^{m+i} \sum_{\ell=1}^{m+i} \left( \left( 8 \left\| \varphi_t^* \right\|_{L^4(P)}^4 + 8 \psi^{*4} \right) \left( 8 \left\| \varphi_{t-i}^* \right\|_{L^4(P)}^4 + 8 \psi^{*4} \right) \right.$$

$$\left. \cdot \left( 8 \left\| \varphi_{t-j}^* \right\|_{L^4(P)}^4 + 8 \psi^{*4} \right) \left( 8 \left\| \varphi_{t-\ell}^* \right\|_{L^4(P)}^4 + 8 \psi^{*4} \right) \right)^{1/2}$$

(for constants $a, b$, it holds $(a + b)^4 \leq 8a^4 + 8b^4$)

$$= o(1). \qquad \qquad \text{(Equation (C.2); there are } O(T) \text{ terms in the sum)}$$

Thus, with Chebyshev's inequality, Equation (C.21) is $o_P(1)$. Then, the fact that Equation (C.20) is $o_P(1)$ and Equation (C.21) is $o_P(1)$ imply that Equation (C.11) is $o_P(1)$. Finally, the fact that Equation (C.10) and Equation (C.11) are each $o_P(1)$ completes the proof of the lemma. $\qquad \square$

## D. Further details and proofs for Section 4

Before we provide an overview of the proofs and prove the results, we state the necessary formal assumptions. In Assumption D.1, we require that each individual has true and predicted probability of assignment to treatment and to control bounded away from zero, almost surely. We also require that the outcome function be bounded in $L^q$ norm, for $q > 4$ and that the error terms $\tilde{Y}_t$ are bounded in $L^2$ norm.

**Assumption D.1** (Regularity conditions for ADE estimation)**.** There exist constants $C, \zeta > 0$ such that the following regularity conditions are met for all $(f, m) \in S_T$ and all $t \in [T]$

$$\zeta < m(X_t) < 1 - \zeta, \text{ a.s.} \tag{D.1}$$

$$\| f(D_t, X_t, H_t) \|_{L^{4+\delta}(P)} < C, \tag{D.2}$$

$$\| \tilde{Y}_t \|_{L^2(P)} < C \tag{D.3}$$

Equation (D.4) says that, on average over units $t = 1, \ldots, T$, the expected squared difference between $m$ and $m^*$ must go

to zero. Equation (D.5) says the same for $f$ and $f^*$. Equation (D.6) says that the average of the product in $m - m^*$ and $f - f^*$ over the data must go to zero at faster-than-$\sqrt{T}$ rates.

**Assumption D.2** (Rate conditions for ADE estimation). *The following rate conditions are met for all $(f, m) \in S_T$*

$$\frac{1}{T} \sum_{t=1}^{T} \|(m^* - m)(X_t)\|_{L^2(P)} = o(1), \tag{D.4}$$

$$\frac{1}{T} \sum_{t=1}^{T} \|(f^* - f)(D_t, X_t, H_t)\|_{L^2(P)} = o(1), \tag{D.5}$$

$$\left| T^{-1} \sum_{t=1}^{T} \mathbb{E}_P \left[ (m - m^*)(X_t) \cdot (f - f^*)(D_t, X_t, H_t) \right] \right| = o_{P_T}(T^{-1/2}), \tag{D.6}$$

In this section, we prove our results for Section 4. The result follows directly from Theorem 3.1 once we have verified the assumptions necessary for the theorem to apply. Thus, our main task is verifying these assumptions.

There are three parts of the assumptions in Section 3 that are not trivially verified by analogous assumptions in Section 4. These are: the Neyman orthogonality of $\psi$, $L^2$ convergence of $\varphi(W_t, \eta)$ to $\varphi(W_t, \eta^*)$ on average over $t$, and a second-order condition stating that the second-order misestimation of $\eta^*$ does not dominate. Each of these is stated in a separate lemma, and we provide high-level descriptions of each of them before they are stated and proved.

*Proof of Theorem 4.1.* If we can verify that Assumptions 2.1, C.2 and C.3 hold, then we can apply Theorem 3.1. Recall that Assumption 2.1 is verified by assumption in the definition of the model in Section 4.1. Assumption C.2(a) is trivially verified from the definition of $\psi$. Assumption C.2(b) is given by Lemma D.3. Assumption C.3, Equation (C.2) is trivially verified by the definition of $\psi$ and Equations (D.1) and (D.2). Assumption C.3, Equation (C.3) is given by Lemma D.4. Assumption C.3, Equation (C.4) is given by Lemma D.5. □

Lemma D.3 states that the estimator $\psi$ must be Neyman orthogonal with respect to the nuisance realization set $S_T$. The argument is standard to any proof of Neyman orthogonality of an augmented inverse probability weighted (AIPW) estimator.

**Lemma D.3** (Neyman orthogonality). *For the model defined in Section 4.1 and the estimand $\psi$ and estimator $\psi^*$ defined in Equations (4.4) and (4.5) respectively, and under Assumptions D.1 and D.2, $\psi(W_{1:T}; \eta)$ is Neyman orthogonal with respect to $S_T$.*

*Proof.* Notice, for all $t$

$$\frac{\partial}{\partial r} \mathbb{E}_P \left[ \varphi(w_t; \eta^* + r(\eta - \eta^*)) \right] \Big|_{r=0}$$

$$= \mathbb{E} \Bigg[ (f - f^*)(1, X_t, H_t) - (f - f^*)(0, X_t, H_t)$$

$$\quad + \left( \frac{D_t}{m^*(X_t)} - \frac{1 - D_t}{1 - m^*(X_t)} \right) (f - f^*)(D_t, X_t, H_t) \Bigg] \quad \text{(Equation (D.2) and the dominated convergence theorem)}$$

$$= \mathbb{E}_P \left[ (f - f^*)(1, X_t, H_t) \left( 1 - \frac{D_t}{m^*(X_t)} \right) - (f - f^*)(0, X_t, H_t) \left( 1 - \frac{1 - D_t}{1 - m^*(X_t)} \right) \right] \quad \text{(Rearranging)}$$

$$= 0. \quad (\mathbb{E}_P \left[ D_t/m^*(X_t) \mid X_t \right] = \mathbb{E}_P \left[ (1 - D_t)/(1 - m^*(X_t)) \mid X_t \right] = 1)$$

□

Lemma D.4 states that, on average over time, $\varphi(W_t; \eta)$ must converge to $\varphi(W_t; \eta^*)$ in $L^2$, uniformly over the nuisance realization set $S_T$. The proof consists of showing that convergence of $\varphi$ reduces to conditions on convergence of $f$ to $f^*$ and $m$ to $m^*$, which we assumed in Assumption D.2.

**Lemma D.4** (Consistency). *For the model defined in Section 4.1 and the estimand $\psi$ and estimator $\psi^*$ defined in Equa-*

*tions* (4.4) *and* (4.5) *respectively, under Assumptions D.1 and D.2,*

$$\sup_{\eta \in S_T} T^{-1} \sum_{t=1}^{T} \|\varphi(W_t; \eta) - \varphi(W_t; \eta^*)\|_{L^2(P)} = o(1).$$

*Proof.* Notice, for $\eta \in S_T$,

$$
\begin{aligned}
&\varphi(W_t; \eta) - \varphi(W_t; \eta^*) \\
&= (f - f^*)(1, X_t, H_t) - (f - f^*)(0, X_t, H_t) \\
&\quad + D_t \left( \frac{Y_t - f(D_t, X_t, H_t)}{m(X_t)} - \frac{Y_t - f^*(D_t, X_t, H_t)}{m^*(X_t)} \right) \\
&\quad - (1 - D_t) \left( \frac{Y_t - f(D_t, X_t, H_t)}{1 - m(X_t)} - \frac{Y_t - f^*(D_t, X_t, H_t)}{1 - m^*(X_t)} \right) \\
&= (f - f^*)(1, X_t, H_t) - (f - f^*)(0, X_t, H_t) \\
&\quad + D_t \left( \frac{(f^* - f)(D_t, X_t, H_t) + \tilde{Y}_t}{m(X_t)} - \frac{\tilde{Y}_t}{m^*(X_t)} \right) \\
&\quad - (1 - D_t) \left( \frac{(f^* - f)(D_t, X_t, H_t) + \tilde{Y}_t}{1 - m(X_t)} - \frac{\tilde{Y}_t}{1 - m^*(X_t)} \right) \qquad \text{(Equation (4.1))} \\
&\leq (f - f^*)(1, X_t, H_t) - (f - f^*)(0, X_t, H_t) \\
&\quad + \zeta^{-2} D_t \left( ((f - f^*)(D_t, X_t, H_t)) \, m^*(X_t) - \tilde{Y}_t((m^* - m)(X_t)) \right) \\
&\quad + \zeta^{-2}(1 - D_t) \left( ((f - f^*)(D_t, X_t, H_t)) (1 - m^*(X_t)) - \tilde{Y}_t((m - m^*)(X_t)) \right) \qquad \text{(Equation (D.1))} \\
&\leq (f - f^*)(1, X_t, H_t) - (f - f^*)(0, X_t, H_t) \\
&\quad + 2\zeta^{-2} \left( ((f - f^*)(D_t, X_t, H_t)) - \tilde{Y}_t((m - m^*)(X_t)) \right) \qquad (D_t, 1 - D_t, m^*(X_t), 1 - m^*(X_t) \text{ are all in } [0, 1])
\end{aligned}
$$

Thus, by the triangle inequality:

$$
\begin{aligned}
&\sum_{t=1}^{T} \|\varphi(W_t; \eta) - \varphi(W_t; \eta^*)\|_{L^2(P)} \\
&= \frac{1}{T} \sum_{t=1}^{T} \|(f - f^*)(1, X_t, H_t)\|_{L^2(P)} \\
&\quad + \|(f - f^*)(1, X_t, H_t)\|_{L^2(P)} \\
&\quad + 2\zeta^{-2} \left( \|(f - f^*)(D_t, X_t, H_t)\|_{L^2(P)} + O_P(1) \|m(X_t) - m^*(X_t)\|_{L^2(P)} \right) \qquad \text{(Equation (D.3))}
\end{aligned}
$$

Next, the assumptions Equation (D.1) and Equation (D.5) combined imply

$$\frac{1}{T} \sum_{t=1}^{T} \|(f - f^*)(1, X_t, H_t)\|_{L^2(P)} = o(1)$$

$$\frac{1}{T} \sum_{t=1}^{T} \|(f - f^*)(0, X_t, H_t)\|_{L^2(P)} = o(1).$$

And finally

$$2\zeta^{-2} \frac{1}{T} \sum_{t=1}^{T} \|(f - f^*)(D_t, X_t, H_t)\|_{L^2(P)} = o(1)$$

$$2\zeta^{-2}\frac{1}{T}\sum_{t=1}^{T}O_P(1)\left\|(m-m^*)(X_t)\right\|_{L^2(P)} = o(1)$$

by Equations (D.4) and (D.5). □

Lemma D.5 states that the second-order term in Taylor's theorem converges to zero at faster-than-$\sqrt{T}$ rates. This ensures we can apply Taylor's theorem and establishes that slow rates of convergence of $\hat{\eta}$ to $\eta^*$ do not lead to sub-optimal rates on convergence of $\psi(W_{1:T},\hat{\eta})$ to $\psi^*$. The proof relies on showing that the second Gateaux derivative of $\psi$ on the path from any $\eta$ in the nuisance realization set $S_T$ to $\eta^*$ reduces to conditions on the *products* of convergence rates of $m$ to $m^*$ and $f$ to $f^*$. Thus, it is sufficient for each of $m$ and $f$ to converge to their true values at faster-than-$T^{1/4}$ rates, or for $m$ to be known *a priori* while $f$ converges at arbitrarily slow rates.

**Lemma D.5** (Second-order condition). *For the model defined in Section 4.1 and the estimand $\psi$ and estimator $\psi^*$ defined in Equations (4.4) and (4.5) respectively, under Assumptions D.1 and D.2,*

$$\sup_{r\in(0,1),\eta\in S_T}\left|\frac{\partial^2}{\partial r^2}\mathbb{E}_P\left[\psi(W_{1:T};\eta^*+r(\eta-\eta^*))\right]\right| = o(T^{-1/2}).$$

*Proof.* Observe, for $\eta \in S_T$ and $r \in (0,1)$,

$$\frac{\partial^2}{\partial r^2}\mathbb{E}_P\left[\varphi(w_t;\eta^*+r(\eta-\eta^*)\right]$$
$$= \mathbb{E}_P\left[-\left(\frac{D_t}{(m^*(X_t)+r((m-m^*)(X_t)))^2}-\frac{1-D_t}{(1-m^*(X_t)-r((m-m^*)(X_t)))^2}\right)\right.$$
$$\left.\cdot(m-m^*)(X_t)\cdot(f-f^*)(D_t,X_t,H_t)\right]$$

where we first switch the derivative with the expectation using Equation (D.2) and the dominated convergence theorem, and then we evaluate the derivatives. This implies

$$\left|\frac{\partial^2}{\partial r^2}\mathbb{E}_P\left[\psi(W_{1:T};\eta^*+r(\eta-\eta^*))\right]\right|$$
$$= \left|T^{-1}\sum_{t=1}^{T}\mathbb{E}_P\left[\frac{D_t}{(m^*(X_t)+r((m-m^*)(X_t)))^2}-\frac{1-D_t}{(1-m^*(X_t)-r((m-m^*)(X_t)))^2}\right.\right.$$
$$\left.\left.\cdot(m-m^*)(X_t)\cdot(f-f^*)(D_t,X_t,H_t)\right]\right|$$
$$\leq \left|T^{-1}\sum_{t=1}^{T}\mathbb{E}[\zeta^{-2}\cdot(m-m^*)(X_t)\cdot(f-f^*)(D_t,X_t,H_t)]\right| \qquad\text{(Equation (D.1))}$$
$$\leq o_P(T^{-1/2}). \qquad\text{(Equation (D.6))}$$

□

# E. Further details and proofs for Section 5

We first state the formal assumptions for inference in Section 5. In Assumption E.1, we require that the probability that any $m+1$ sequential treatment assignments are the same (either all treatment or all control) is bounded away from zero. We also require that the conditional expectation estimator $f$ and the true conditional expectation function $f^*$ be bounded in $L^q$ norm for $q > 4$. In Assumption E.2, we require that the time-average of the $L^2$ norm of $f-f^*$ is going to zero as $T \to \infty$.

**Assumption E.1** (Regularity conditions for GATE estimation in switchback experiments). *There exist constants $C, \zeta > 0$ such that the following regularity conditions are met for all $t \in [T]$*

$$\pi^*((t-m):t,\mathbf{d}) \geq \zeta,\ \text{a.s.} \qquad\qquad \forall \mathbf{d}\in\{\mathbf{0},\mathbf{1}\} \qquad\qquad \text{(E.1)}$$
$$\|f(D_t,X_t,H_t)\|_{L^{4+\delta(P)}} < C \qquad\qquad\qquad\qquad\qquad\quad \text{(E.2)}$$
$$\|f^*(D_t,X_t,H_t)\|_{L^{4+\delta(P)}} < C \qquad\qquad\qquad\qquad\qquad\quad \text{(E.3)}$$

**Assumption E.2** (Rate conditions for GATE estimation in switchback experiments). The following rate conditions are met:

$$T^{-1}\sum_{t=1}^{T}\|(f - f^*)(D_t, X_t, H_t)\|_{L^2(P)} = o(1) \tag{E.4}$$

In this section, we provide proofs for our results in Section 5. The proof of Theorem 5.1 mirrors that of Theorem 4.1: we just need to verify the assumptions necessary for applying Theorem 3.1. This again reduces to verifying a Neyman orthogonality condition, a consistency condition and a second-order Gateaux derivative condition.

*Proof of Theorem 5.1.* As in the proof of Theorem 4.1, we just need to verify Assumption C.2 and Assumption C.3 and apply Theorem 3.1. Assumption C.2(a) is trivially verified by the definition of $\psi$. Assumption C.2(b) is verified by Lemma E.3. For Assumption C.3, Equation (C.2) is verified by Equations (E.1) to (E.3) and the definition of $\varphi$. Equation (C.3) is verified by Lemma E.4. Equation (C.4) is verified by Lemma E.5. □

Lemma E.3 states that our estimator is Neyman orthogonal with respect to the nuisance realization set. The proof resembles standard arguments of AIPW estimators, except that in our case the inverse-propensity is the replaced with the probabilities that the last $m$ observations where all assigned to treatment or control.

**Lemma E.3** (Neyman orthogonality). *For the model defined in Section 5.1, and the estimand and estimator defined in Equations (5.4) and (5.5), under Assumptions E.1 and E.2, then $\psi(W_{1:T}; \eta)$ is Neyman orthogonal with respect to $S_T$.*

*Proof of Lemma E.3.* Notice:

$$\frac{\partial}{\partial r}\mathbb{E}\big[\varphi(W_t; \eta^* + r(\eta - \eta^*))\big]\big|_{r=0}$$
$$= \mathbb{E}\left[(f - f^*)(1, X_t, H_t(\mathbf{1})) - (f - f^*)(0, X_t, H_t(\mathbf{0}))\right]$$
$$\left(\frac{\mathbb{1}\left\{D_{(t-m):t} = \mathbf{1}\right\}}{\pi^*((t-m):t, \mathbf{1})} - \frac{\mathbb{1}\left\{D_{(t-m):t} = \mathbf{0}\right\}}{\pi^*((t-m):t, \mathbf{0})}\right)(\tilde{Y}_t + (f - f^*)(D_t, X_t, H_t)]$$
$$= \mathbb{E}\Big[(f - f^*)(1, X_t, H_t(\mathbf{1}))\left(1 - \frac{\mathbb{1}\left\{D_{(t-m):t} = \mathbf{1}\right\}}{\pi^*((t-m):t, \mathbf{1})}\right)$$
$$- (f - f^*)(0, X_t, H_t(\mathbf{0}))\left(1 - \frac{\mathbb{1}\left\{D_{(t-m):t} = \mathbf{0}\right\}}{\pi^*((t-m):t, \mathbf{0})}\right)\Big]$$

where the first equality follows from Equations (E.1) to (E.3) and applying the dominated convergence theorem. The second equality follows by the fact that $\mathbb{E}\left[\tilde{Y}_t \mid D_{(t-m):t}\right] = 0$ and rearranging. Now, using the facts that

$$\mathbb{E}_P\left[\mathbb{1}\left\{D_{(t-m):t} = \mathbf{1}\right\} \mid X_{(t-m):t}\right] = \mathbb{E}_P\left[\mathbb{1}\left\{D_{(t-m):t} = \mathbf{1}\right\}\right] = \pi^*((t-m):t, \mathbf{1}), \text{ and}$$
$$\mathbb{E}_P\left[\mathbb{1}\left\{D_{(t-m):t} = \mathbf{0}\right\} \mid X_{(t-m):t}\right] = \mathbb{E}_P\left[\mathbb{1}\left\{D_{(t-m):t} = \mathbf{0}\right\}\right] = \pi^*((t-m):t, \mathbf{0}),$$

and iterated expectations, the Gateaux derivative is zero. □

Lemma E.4 states the $L^2$ convergence of $\varphi(W_t; \eta)$, uniformly over the nuisance realization set $S_T$, to $\varphi(W_t; \eta^*)$. Similar to the previous section, verifying the result reduces to conditions on the convergence of the nuisances, which in this case is satisfied by the a requirement that $f$ converges to $f^*$ and treatment probabilities are known.

**Lemma E.4** (Consistency). *For the model defined in Section 5.1, and the estimand and estimator defined in Equations (5.4) and (5.5), under Assumptions E.1 and E.2, then*

$$\sup_{\eta \in S_T} T^{-1}\sum_{t=1}^{T}\|\varphi(W_t; \eta) - \varphi(W_t; \eta^*)\|_{L^2(P_T)} = o(1).$$

*Proof of Lemma E.4* Notice

$$\varphi(W_t; \eta) - \varphi(W_t; \eta^*) = (f - f^*)(1, X_t, H(\mathbf{1})) - (f - f^*)(0, X_t, H(\mathbf{0}))$$

$$+ \left( \frac{\mathbb{1}\left\{D_{(t-m):t} = \mathbf{1}\right\}}{\pi^*((t-m):t, \mathbf{1})} - \frac{\mathbb{1}\left\{D_{(t-m):t} = \mathbf{0}\right\}}{\pi^*((t-m):t, \mathbf{0})} \right) (f - f^*)(D_t, X_t, H_t).$$

Now, since, by Equation (E.1) and Equation (E.4), it holds

$$\|(f - f^*)(\mathbf{b}, X_t, H(\mathbf{b}))\|_{L^2(P)} = o(1), \qquad\qquad \forall \mathbf{b} \in \{\mathbf{0}, \mathbf{1}\}$$

and by the fact that

$$\left| \frac{\mathbb{1}\left\{D_{(t-m):t} = \mathbf{1}\right\}}{\pi^*((t-m):t, \mathbf{1})} - \frac{\mathbb{1}\left\{D_{(t-m):t} = \mathbf{0}\right\}}{\pi^*((t-m):t, \mathbf{0})} \right| \leq \max\left\{\pi^*((t-m):t, \mathbf{1})^{-1}, \pi^*((t-m):t, \mathbf{0})^{-1}\right\} \leq \zeta^{-1}$$

almost surely, the triangle inequality proves the result. $\qquad\square$

Lemma E.5 states the condition that second-order terms on the path from $\eta$ in $S_T$ to $\eta^*$ do not dominate. In this case, the condition is easily verified by the fact that the setting is experimental and therefore treatment probabilities are known.

**Lemma E.5** (Second-order condition). *For the model defined in Section 5.1, and the estimand and estimator defined in Equations* (5.4) *and* (5.5)*, under Assumptions E.1 and E.2, then*

$$\sup_{r \in (0,1), \eta \in S_T} \left| \frac{\partial^2}{\partial r^2} \mathbb{E}_P \left[ \psi(W_{1:T}; \eta^* + r(\eta - \eta^*)) \right] \right| = o(T^{-1/2}).$$

*Proof of Lemma E.5.* The result holds trivially by Equations (E.1) to (E.3) and applying the dominated convergence theorem (to switch the derivative and expectation) and the fact that the second derivative is 0. $\qquad\square$

# F. Additional simulation details

## F.1. Simulations for Section 4

For each simulation, we set $T = 1{,}000$ and $d_X = 10$. For all $i \in 1, \ldots, d_X$, we let $X_{t,i} \sim N(1,1)$ iid. $D_t \sim \mathrm{Ber}(\min\{\max\{\zeta, X_{t,1}\}, 1 - \zeta\})$ and $\zeta = 0.1$. The outcome function is defined as

$$Y_t(D_t, H_t) \stackrel{\text{def.}}{=} \sin(2\pi X_{t,1}) + 2D_t + 2H_t D_t - 1 + \tilde{Y}_t,$$

$$H_t(W_{t-1}) \stackrel{\text{def.}}{=} 0.75(H_{t-1} - 1) + 1 + \tilde{H}_t.$$

where $\tilde{Y}_t \sim \mathcal{N}(0, 1/10)$, $\tilde{H}_t \sim \mathcal{N}(0,1)$ iid. For Figure 2, we run 50,000 simulations. In Figure 5, we run 1,000 simulations for each time horizon $T$ and plot 10 values of $T$ evenly divided on a logarithmic scale from 100 to 10,000.

In Figure 5, for each time horizon $T$, we compute 1,000 simulations. We also compute the empirical standard error for each estimate in ribbons around the point estimates by computing the empirical standard deviation of the point estimates across simulations and dividing by the square root of the number of simulations. We next provide additional comparisons between confidence intervals constructed via our estimator and other estimators.

**Estimators for treatment effect comparisons.** The naive plug-in estimator is defined formally as

$$\psi^{\mathrm{pi}}(W_{1:T}; \eta) \stackrel{\text{def.}}{=} T^{-1} \sum_{t=1} \varphi^{\mathrm{pi}}(W_t; \eta) \tag{F.1}$$

where $\varphi^{\mathrm{pi}}$ is defined as in Equation (4.5).

The naive Horvitz-Thompson is defined formally as

$$\psi^{\mathrm{HT}}(W_{1:T}; \eta) \stackrel{\text{def.}}{=} T^{-1} \sum_{t=1}^{T} \left( \frac{D_t}{m(X_t)} - \frac{1 - D_t}{1 - m(X_t)} \right) Y_t. \tag{F.2}$$

The naive DML estimator is defined formally as

$$\psi^{\mathrm{DML-N}}(W_{1:T}; \eta') \overset{\text{def.}}{=} T^{-1} \sum_{t=1}^{T} \varphi^{\mathrm{DML-N}}(W_t; \eta') \tag{F.3}$$

where

$$
\begin{aligned}
\varphi^{\mathrm{DML-N}}(W_t; \eta') \overset{\text{def.}}{=} & f'(1, X_t) - f'(0, X_t) \\
& + \left( \frac{D_t}{\hat{m}(X_t)} - \frac{1 - D_t}{1 - \hat{m}(X_t)} \right) \cdot (Y_t - f'(D_t, X_t)).
\end{aligned}
$$

The naive DML estimator $\psi^{\mathrm{DML-N}}$ is simply the canonical iid AIPW estimator, ignoring the shared state. The nuisances are $\eta' = (f', \hat{m})$, where we learn an outcome model $f'$ by learning a predictor of $Y_t$ given $X_t$ and $D_t$ but omitting the shared state, and learn $\hat{m}$ as before by predicting $D_t$ as a function of $X_t$. (We use the notation $f'$ to disambiguate from $\hat{f}$, which takes $H_t$ as an argument.) To learn $f'$, we use an auxiliary sample as in Algorithm 1.

**Setup for confidence interval comparisons.** We next investigate the coverage of confidence intervals constructed from treatment effect and variance estimators. Confidence intervals for an arbitrary pair of treatment effect, variance estimators $(\hat{\psi}, \hat{\sigma}^2)$ are constructed as

$$\mathrm{CI}_\alpha(\hat{\psi}, \hat{\sigma}^2) = [\hat{\psi} - z_\alpha \hat{\sigma}/\sqrt{T}, \hat{\psi} + z_\alpha \hat{\sigma}/\sqrt{T}]$$

where $z_\alpha$ is the $(1 - \alpha/2)$-th quantile of a standard normal distribution. To compute coverage rates, we run many simulations, constructing a confidence interval for each one and calculating the proportion of confidence intervals that contain the estimand $\psi^*$.

The DML4SSI confidence intervals are constructed using $\psi(W_{1:T}, \hat{\eta})$ and the consistent variance estimator $\hat{\sigma}^2$ in Theorem 3.1. The Horvitz-Thompson (HT) confidence intervals use the treatment effect estimator $\psi^{\mathrm{HT}}(W_{1:T}; \hat{\eta})$ as defined in Equation (F.2) and a standard variance estimator $\hat{\sigma}^2_{\mathrm{HT}}$ defined as

$$
\begin{aligned}
\hat{\sigma}^2_{\mathrm{HT}} = T^{-1} \sum_{t=1}^{T} \Bigg( & \frac{D_t}{\hat{m}(X_t)} (Y_t - \overline{Y}^{(0)}) \\
& - \frac{1 - D_t}{1 - \hat{m}(X_t)} (Y_t - \overline{Y}^{(1)}) \Bigg)^2
\end{aligned}
$$

where $\overline{Y}^{(\ell)}$ for $\ell \in \{0, 1\}$ is the mean observed outcome conditional on $D_t = \ell$. The plug-in confidence intervals use $\psi^{\mathrm{pi}}(W_{1:T}; \eta)$ as defined in Equation (F.1) and the variance estimator defined in Theorem 3.1, substituting $\varphi^{\mathrm{pi}}$ for $\varphi$ and $\psi^{\mathrm{pi}}$ for $\psi$. The naive DML (DML-N) estimator uses the treatment effect estimator $\psi^{\mathrm{DML-N}}$ defined as

$$
\begin{aligned}
\hat{\sigma}^2_{\mathrm{DML-N}} = T^{-1} \sum_{t=1}^{T} \big( & \varphi^{\mathrm{DML-N}}(W_t; \hat{\eta}) \\
& - \psi^{\mathrm{DML-N}}(W_{1:T}; \hat{\eta}) \big)^2.
\end{aligned} \tag{F.4}
$$

In addition to the HT, plug-in and DML-N estimators compared above, we also compare our method to one where we treat the shared-state as covariates and construct treatment effect and variance estimates accordingly. We call these the *shared-state as covariates (SSAC)* estimators. In this section, the SSAC treatment effect estimator is mechanically the same as our treatment effect estimator, since both of them involve learning a conditional expectation function estimate $\hat{f}$. (In Section 5, this is not true, and our treatment effect estimator and the SSAC estimators will be different, leading to inconsistent SSAC treatment effect estimates.) However, the SSAC variance estimator ignores covariances between terms and treats the data as if there were no covariances between terms. This is different from our variance estimator, which does account for covariances between terms. The SSAC variance estimator $\hat{\sigma}^2_{\mathrm{SSAC}}$ is defined as in Equation (F.4), substituting $\varphi$ for $\varphi^{\mathrm{DML-N}}$ and $\psi$ for $\psi^{\mathrm{DML-N}}$. These are the variance estimates that would have been constructed had the shared states been treated like iid covariates.

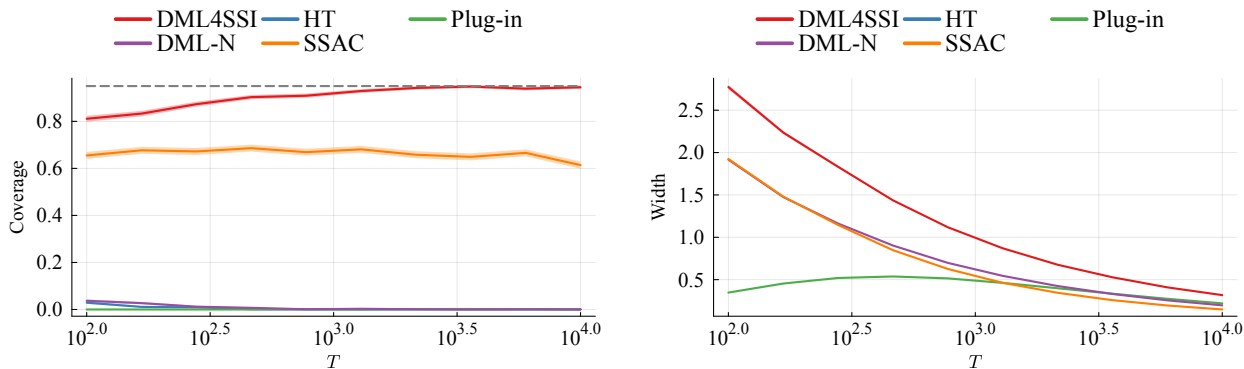

**Figure 5:** Coverage rates of 95% confidence intervals for the ADE constructed using our estimators (DML4SSI), shared state as covariates (SSAC) estimators, naive Horvitz-Thompson estimators, plug-in estimators and naive DML estimators. The HT, plug-in and naive DML estimators coverage overlap around 0. We plot standard errors of the simulations in lighter-colored ribbons around the coverage point estimates.

**Results for confidence interval comparisons.** In Figure 5, we run simulations for different values of $T$ and construct confidence intervals for each estimator. In the left plot, we show coverage rates of the confidence intervals, and in the right plot, we show the confidence interval widths. We observe that the DML4SSI confidence intervals approach the target 95% coverage as $T$ increases, while each of the other estimators fall substantially below the target coverage. In particular, each of the HT, plug-in and DML-N confidence are substantially biased as we saw in Figure 2, leading to confidence intervals with coverage close to zero. The SSAC confidence intervals use the same (consistent) treatment effect estimator as the DML4SSI but have variance estimates that do not account for covariance between observations over time. This leads to underestimates of the variance, confidence intervals that are too narrow and lower-than-desired coverage. In this case, the SSAC 95% confidence intervals result in coverage between 0.6 and 0.7, and do not seem to be converging to the target coverage as $T$ increases.

### F.2. Simulations for Section 5

We let $d_X = 1$ and $X_t \sim \text{Unif}([0, 1])$. Also, we let $D_t \sim \text{Ber}(\min\{\max\{\zeta, X_{t,1}\}, 1 - \zeta\})$ where $\zeta = 0.1$. We let $H_t$ be the last $m$ observations $X_{t-m:t-1}, D_{t-m:t-1}$ where $m = 5$. We let

$$Y_t(D_t, H_t) \overset{\text{def.}}{=} \sin(2\pi X_{t,1}) + 2D_t + 2 \sum_{i \in \lfloor m/6 \rfloor} e^{-H_{t,6i}/3} - 1 + \tilde{Y}_t,$$

where $\tilde{Y}_t \sim N(0, 1/10)$ iid. We set the switching period to $2m$.

In our simulations, we let $T = 1{,}000$. In Figures 3 and 4, we run each simulation 50,000 times. In Figure 6, for each time horizon $T$, we run 1,000 simulations.

**Estimators for treatment effect comparisons.** The shared-state as covariates (SSAC) estimator is defined formally as:

$$\psi^{\text{SSAC}}(W_{1:T}; \eta) \overset{\text{def.}}{=} T^{-1} \sum_{t=1}^{T} f(1, X_t, H_t) - f(0, X_t, H_t) \tag{F.5}$$
$$+ \left( \frac{D_t}{m(X_t)} - \frac{1 - D_t}{1 - m(X_t)} \right) \cdot (Y_t - f(D_t, X_t, H_t)).$$

The unbiased switchback estimator is defined as

$$\psi^{\text{SB}}(W_{1:T}) \overset{\text{def.}}{=} \sum_{t=1}^{T} \left( \frac{\mathbb{1}\{D_{(t-m):t} = \mathbf{1}\}}{\pi^*((t-m):t, \mathbf{1})} - \frac{\mathbb{1}\{D_{(t-m):t} = \mathbf{0}\}}{\pi^*((t-m):t, \mathbf{0})} \right) Y_t. \tag{F.6}$$

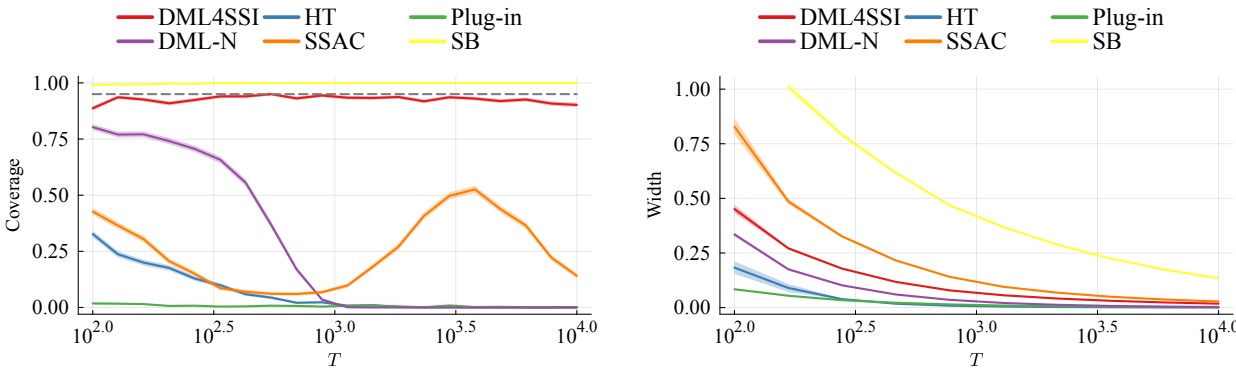

**Figure 6:** Coverage rates and widths of 95% confidence intervals for the GATE constructed using our estimators (DML4SSI), switchback Horvitz-Thompson (HT) estimators, naive HT estimators, plug-in estimators and naive DML estimators. In the left plot, the coverage for confidence intervals constructed using the naive HT, plug-in and naive DML estimators overlap around 0. We plot standard errors of the simulations in lighter-colored ribbons around the coverage point estimates.

**Confidence interval comparisons.**   We next explore the coverage of confidence intervals constructed from treatment effect and variance estimators. We use the same procedure to generate confidence interval coverage Figure 6 as we do for Figure 5: we run many simulations where we generate confidence intervals and compute the fraction of them that contain the true effect $\psi^*$. The definitions of the treatment effect and variance estimators for the naive HT, plug-in, naive DML estimators are the same as in Section 4.3. We use our DML treatment effect estimator $\psi(W_{1:T}; \hat{\eta})$ specified by Equation (5.5) and the consistent variance estimator from Theorem 3.1. For the switchback HT estimators, we use the treatment effect estimator from Equation (F.6) and the conservative variance estimator provided in Bojinov et al. (2022, Corollary 1). For the SSAC estimators, we use the treatment effect estimator from Equation (F.5) and the naive variance estimator as defined in Section 4.3.

We observe that the DML4SSI confidence intervals are close to the target 95% coverage, while the switchback HT estimator provides substantially greater than 95% coverage. This is consistent with the fact that their variance estimator is conservative and will not be consistent in general. The mean confidence interval width constructed from the SB estimator (yellow) is between 3 and 7.5 times the width of that constructed from the DML4SSI estimator (red): put another way, in these simulations, the DML4SSI estimator requires many fewer samples to produce confidence intervals the same width as the SB estimator for a given time horizon $T$. Finally, we note that the naive estimators and plugin estimators all provide coverage near zero. This occurs as a result of the biases of each of these estimators for estimation of the treatment effect.

