# OpenReview forum: "Double Machine Learning for Causal Inference under Shared-State Interference"
_ICML.cc/2025/Conference — ICML 2025 poster_

### Official Review · Reviewer_wSH2 · 2025-03-11

**Overall Recommendation:** 4

**Summary:**

This paper unifies the set of problems in causal inference with interference where the outcomes of individuals depend on others' treatment assignment only through an observed shared state. The paper assumes the units arrive sequentially and models this shared-state problem using the Markov chain. The paper then proposes a double robust machine learning meta-estimator to estimate causal quantities of interest and proves the estimator's asymptotic properties. The paper shows that both average direct effects and global average treatment effects can be estimated by this meta-estimator. The paper conducts numerical experiments to demonstrate the effectiveness of the proposed estimator.

## update after rebuttal
The authors have addressed my concern. I will retain my score.

**Claims And Evidence:**

On line 83, could the authors explain why it is realistic to assume that the covariate X_t is independent of the hidden state H_t, perhaps using the example described in the introduction section? Is there a way to relax this? This also applies to invariance of the conditional distribution of H_t and D_t.

On line 182, the authors assume that the nuisance estimators can be generated via an auxiliary sample of data. In reality, when is this auxiliary sample of data available?

**Essential References Not Discussed:**

Could the authors discuss more about how the prior works in various fields relate to the concept of "shared-state interference", and how the paper unifies this concept ( perhaps In the appendix)?

**Experimental Designs Or Analyses:**

Yes, I did. Since the authors have derived the asymptotic normality for the estimator, they shall also report the actual coverage of the estimator in the experiment.

**Methods And Evaluation Criteria:**

Yes, they make sense.

**Other Comments Or Suggestions:**

line 82: "independent of the indentities of..."

line 85: "...develops a DML theorem a discrete choice model..."

line 151: "Hence, they are nuisances." is redundant

line 307: "Next, we our validate our results"

line 316: "averate treatment effect"

The word "adjustment" only appears once in the title of section 5. Consider to replace the title with "Inference on Global Average Treatment Effect "

**Other Strengths And Weaknesses:**

It is a theoretically rigorous paper with significant practical implications. The assumptions made are generally reasonable.

**Questions For Authors:**

Is this the first paper to propose the concept of shared-state interference? Which existing papers is the most closely related to this one?

**Relation To Broader Scientific Literature:**

The problem studied in this paper appears to unify applications across various fields, as suggested by the authors in the introduction. However, the paper does not include an analysis of a real-world dataset. While the oracle causal effect is elusive in real data, it would still be beneficial for the authors to demonstrate their method on at least one such dataset. Additionally, I am curious to see how the authors justify their assumptions and the existence of auxiliary data used for estimating the nuisance parameter in a real-data setting.

If the authors do this, I will raise my score.

**Theoretical Claims:**

Yes, I checked the correctness of the proofs.

---

> ### Author Rebuttal · Authors · 2025-03-30
>
> We thank the reviewer for their thoughtful feedback. We will incorporate your suggestions, including providing coverage rates in Appendix F, in our updated manuscript. In the coverage rate plots, our consistent variance estimator approaches the target coverage rate as $T$ grows, while the coverage rates for naive estimators is near zero due to their bias.
> * **Dependence of $X_t$ and $D_t$ on $H_t$.** (Copying this from our response to reviewer c4Lj.)  This problem formulation would indeed be more general and would be a valuable direction for future work. We made the choice to have $H_t$ affect $Y_t$ but not $X_t$ and $D_t$ for two reasons. First, we found it hard to imagine situations in which $H_t$ exerts significant influence on unit characteristics and treatment assignments: most often, unit characteristics should be innate to the unit (e.g. their price sensitivity or preferences for particular types of content) and treatment assignments should be exogenous (we imagine treatment conditions like new features on a platform or inducements like discounts). Second, our context presents the simplest possible change from canonical iid models and so our work can be directly instructive for comparison with models where no shared-state is present by simply removing the shared-state variable from the outcome structural equations. Assuming dependence of $X_t$ and $D_t$ on $H_t$ would be possible under Theorem 3.1 as long as the data $W_t$ still obeys geometric ergodicity and detailed balance.
> * **Existence of auxiliary data.** Auxiliary data may exist when there are multiple similar systems or markets where unit behavior will be similar. For example, on a social media platform with multiple forums, the data from one forum might be used to construct nuisance estimates for measuring treatment effects in another forum. This is an assumption common to other methods using machine learning for inference or uncertainty quantification (see, e.g., Angelopoulos et al 2023), but of course, this is a limitation on how widely applicable any such method requiring auxiliary data can be. On the other hand, conceptually, the auxiliary sample assumption cleanly separates the machine learning from application of the learned predictors for inference. Relaxations of this requirement may be possible, but we wanted to preserve the conceptual clarity in our paper, so we leave these extensions for future work.
> * **Real-world validations.** Analysis of real-world settings would be a valuable contribution for future work. We wanted to include simulations for the sake of simplicity and brevity.
> * **Related work.** We are not aware of other methods that formalize causal inference through shared-state interference as we do. Several other papers formalize interference through markets or through recommender systems as we note in the related work, but these are context-specific. For example, Munro 2024 allows for causal inference under shared-state interference in settings like auctions where the shared-states are prices and allocations of goods exhibit a cutoff structure. Our work is complementary by offering a context-agnostic approach that does not require, e.g., knowledge of the allocation mechanism if the data generating process satisfies our Markov chain assumption.
>
>
> A. N. Angelopoulos, S. Bates, C. Fannjiang, M. I. Jordan, and T. Zrnic. Prediction-Powered Inference, Nov. 2023. URL http://arxiv.org/abs/2301.09633. arXiv:2301.09633 [cs, q-bio, stat].
>
> E. Munro. Causal Inference under Interference through Designed Markets. 2024.

---

> > ### Comment · Reviewer_wSH2 · 2025-04-02
> >
> > I thank the authors for addressing my questions and responding to my comments. I suggest clearly indicating the 95% threshold on the Y-axis.

---

> > > ### Author Response · Authors · 2025-04-03
> > >
> > > As far as we are aware, we aren't allowed to update the pdf after submission. See: https://icml.cc/Conferences/2025/PeerReviewFAQ
> > >
> > > According to the same page, we *are* allowed to share links. The figures we will add along with the corresponding explanatory text are available at the following link: https://docs.google.com/document/d/e/2PACX-1vRiWR2njelj5qIPR8LseWr2gJkLetrSuTK4ks_i2PMaVBfgtTn1zXQ9BY9jQR_Uzo5WkYGam9E7dm66/pub

---

### Official Review · Reviewer_s6JS · 2025-03-12

**Overall Recommendation:** 3

**Summary:**

This paper addresses causal inference under unit interference in systems like markets and recommendation platforms. To model inter-individual interference without strong assumptions, the authors introduce shared-state variables, assuming individual outcomes depend on others only through these shared states. The authors apply double machine learning (DML) for efficient inference, using an auxiliary data sample for nuisance estimators instead of cross-fitting, to account for the sequential nature of sampling. The methodology is used to estimate the average direct effect (ADE) and the global average treatment effect (GATE), with simulations showing DML’s advantages in debiasing and variance reduction.

**Claims And Evidence:**

Yes.

**Essential References Not Discussed:**

I believe this paper includes the vast majority of essential references.

**Experimental Designs Or Analyses:**

This paper presents simulation results for estimating the average direct effect (ADE) and the global average treatment effect (GATE). However, the simulation settings—particularly the one-dimensional covariance structure—seem somewhat simplistic. Exploring more complex scenarios would enhance the generalizability of the results. Additionally, while the evaluation focuses on the magnitude of bias and variance, it overlooks an examination of the consistency of variance estimation. Including the coverage rate of parameter estimates as an additional metric could provide valuable supporting evidence.

**Methods And Evaluation Criteria:**

Yes.

**Other Comments Or Suggestions:**

No.

**Other Strengths And Weaknesses:**

Strengths: This paper clearly introduces the two core aspects of DML: Neyman orthogonality and cross-fitting. It also points out the difference between share-state interference and the traditional i.i.d. data generation mechanism, thus requiring additional modifications to the DML procedure.

Weaknesses: The notation in some places of this paper needs to be consistent. For example, in Section 4.1, Equation 4.1 is written as $Y_t(D_t, H_t)$, while Equation 4.5 uses $Y_t(D_t, X_t, H_t)$.

**Questions For Authors:**

No.

**Relation To Broader Scientific Literature:**

This paper combines shared-state interference with the DML method, both of which have been extensively discussed in the causal inference literature. A related work, *Causal Inference under Interference through Designed Markets*, applies localized debiased machine learning (LDML) for causal inference under shared-state interference in a two-sided market. In this context, the current paper extends the application of DML to causal inference in interference scenarios.

**Theoretical Claims:**

Yes, the proofs for the theorems on the asymptotic properties of DML appear to be largely correct.

---

> ### Author Rebuttal · Authors · 2025-03-30
>
> We thank the reviewer for their comments. We will incorporate this feedback in our updated manuscript.
>
> * **Application-based simulations.** Exploration of more complex and real-world scenarios (using either simulated or real data) would be a valuable direction for future work. Our focus in this paper is on the novel theoretical methodological contribution. Our simulation settings were intended to be simple, to demonstrate how naive estimates can be biased even in simple, one-dimensional settings. Thank you for the suggestion to add coverage rates of variance estimates. We will include this in the appendix.
> * **Notation.** Thank you for pointing out the notational inconsistencies. We will fix these and adhere to potential outcome notation where the dependence on $X_t$ is implicit.

---

### Official Review · Reviewer_c4Lj · 2025-03-18

**Overall Recommendation:** 4

**Summary:**

This paper introduces a double machine learning (DML) estimator for sequentially collected samples where dependencies follow a Markovian structure through a shared-state variable $H_t$.

**Claims And Evidence:**

1. The paper has a strong theorical foundation.
2. Empirical simulations are lacking of specific explanations (e.g., the data generating processes). No empirical evidence for the fast convergence (debiasedness) and doubly robustness are provided.
3. Intuitive explanations on Assumptions C1 and C2 are required for assessing the claims, since they are key assumptions in the paper.
4. Even if it's mentioned that HT estimator is unbiased for estimating GATE, it seems that the HT estimator is biased in Figure 2. Could you please explain the simulation detail?

**Essential References Not Discussed:**

N/A

**Experimental Designs Or Analyses:**

1. Empirical simulations are lacking of specific explanations (e.g., the data generating processes), so it's hard to assess.
2. Even if it's mentioned that HT estimator is unbiased for estimating GATE, it seems that the HT estimator is biased in Figure 2. Could you please explain the simulation detail?

**Methods And Evaluation Criteria:**

1. Empirical simulations are lacking of specific explanations (e.g., the data generating processes), so it's hard to assess.
2. It would be great if the experiments are done with real-world examples.

**Other Comments Or Suggestions:**

1. I think $Y_t(D_t, H_t) = f^*(D_t, X_t, H_t) + \tilde{Y}_t,$ is a wrong formulation. $Y_t$ should be dependent on $X_t$.
2. $D_t$ is not affected by $H_{t-1}$? This doesn’t make sense.
3. Is W_{aux} is iid, while W1,…,WT are dependent in Markovian?
4. Why Assumption C.2 has been made with high-probability? I meant, can we just assume with $\gamma = 0$?

**Other Strengths And Weaknesses:**

N/A

**Questions For Authors:**

-

**Relation To Broader Scientific Literature:**

This paper tackles an interesting problem, where samples are accumulated in Markovian sense. However, I think the paper makes pretty strong and unrealistic assumptions such that Ht only affects Yt, not Xt and Dt. If you assumed so, what are the most difficult challenges?

**Theoretical Claims:**

1. Theoretical results are sound.

---

> ### Author Rebuttal · Authors · 2025-03-30
>
> We thank the reviewer for their feedback, as well as their comments about the soundness of theoretical results and strength of foundation for the work. We will incorporate your comments into our work. We clarify several points and answer questions below.
> * **Data generating process details.** We provide specific explanations of the data generating process in Appendix F, as we note in Line 300, column 2.
> * **Naive Horvitz-Thompson (HT) versus switchback HT estimators.** The HT estimator in figure 2 is a naive HT estimator which merely takes a difference in means across treatment and control observations. The SB estimator in figure 3 is a different estimator which averages only over observations where the last $m$ have been all treatment or all control. The naive HT estimator, in general, may be biased since it does not account for the shared state, but the SB estimator is unbiased. We will clarify this distinction, especially since the SB estimator is a HT-style estimator but is different from the naive HT estimator.
> * **$H_t$ only affects $Y_t$, not $X_t$ and $D_t$.** This problem formulation would indeed be more general and would be a valuable direction for future work. We made the choice to have $H_t$ affect $Y_t$ but not $X_t$ and $D_t$ for two reasons. First, we found it hard to imagine situations in which $H_t$ exerts significant influence on unit characteristics and treatment assignments: most often, unit characteristics should be innate to the unit (e.g. their price sensitivity or preferences for particular types of content) and treatment assignments should be exogenous (we imagine treatment conditions like new features on a platform or inducements like discounts). Second, our context presents the simplest possible change from canonical iid models and so our work can be directly instructive for comparison with models where no shared-state is present by simply removing the shared-state variable from the outcome structural equations. Assuming dependence of $X_t$ and $D_t$ on $H_t$ would be possible under Theorem 3.1 as long as the data $W_t$ still obeys geometric ergodicity and detailed balance.
> * **Notation.** We use standard potential outcome notation with exposure mapping where potential outcomes are typically not written as a function of covariates (even though outcomes depend on covariates). In other words, $Y_t$ *does* depend on $X_t$, but this dependence is omitted from notation. We note that we inadvertently included $X_t$ in the potential outcome notation of some equations, like 4.5, and we will fix this. Thanks for the question.
> * **Independence of $W_{aux}$.** $W_{aux}$ need not be iid as long as the nuisance parameter learners converge at appropriate rates. The tradeoff between quality (say, independence) and quantity (number of observations) of data is implicit in the nuisance learner rates assumptions, but exploration of this question would be a valuable direction for future work.
> * **Choices of $\gamma$.** You can assume $\gamma = 0$ as a special case of Theorem 3.1. We include the high-probability bounds because it follows the results in Chernozhoukov and because it allows for the possibility that eta falls outside of the nuisance realization set with some probability. $\gamma$ might be useful if, as we note in our response to Reviewer YLJC, $m$ may be mis-specified with some probability.

---

### Official Review · Reviewer_YLJC · 2025-03-21

**Overall Recommendation:** 3

**Summary:**

This paper studies causal inference in the presence of *shared-state interference*, a common structure in real-world systems such as online marketplaces and recommender platforms, where outcomes of individuals are influenced by a low-dimensional global variable (e.g., price, inventory, recommendations). The authors formalize this structure and develop a general semiparametric framework under which treatment effects can still be estimated efficiently using **Double Machine Learning (DML)**. The core contribution is an extension of the DML theorem (Chernozhukov et al., 2018) to settings where units arrive sequentially and interfere through a shared Markovian state. The paper derives asymptotic normality and consistent variance estimators for two estimands: the **Average Direct Effect (ADE)** and the **Global Average Treatment Effect (GATE)**, using switchback experiments. Simulations confirm the theoretical properties and demonstrate that the proposed estimators outperform naive plug-in and Horvitz-Thompson alternatives.

**Claims And Evidence:**

The paper claims that:
- Causal inference under shared-state interference can be handled with an appropriately extended DML framework.
- Under certain assumptions (Markovian dynamics, ergodicity, sample splitting or auxiliary data), valid and efficient estimation of ADE and GATE is possible.
- The proposed DML estimators outperform naive baselines in both bias and variance.

These claims are well-supported:
- The extension of the DML theorem is rigorous and builds on a solid foundation.
- The authors provide precise conditions and detailed proofs for asymptotic normality and consistency.
- Simulations effectively illustrate both downward and upward biases in naive estimators, and the superior performance of the DML estimators.

**Essential References Not Discussed:**

No critical omissions were found. The paper cites essential works in DML, interference modeling, ergodic Markov chains, and semiparametric inference. It also offers comparison with network-based approaches, positioning its contribution as orthogonal and novel.

**Experimental Designs Or Analyses:**

The simulations are thoughtfully designed:
- Two estimands (ADE and GATE) are estimated across multiple conditions.
- Competing estimators (plug-in, Horvitz-Thompson) are included for comparison.
- The authors use random forests as flexible nuisance estimators, with auxiliary datasets to avoid sample leakage.

Figures clearly show the bias and variance behavior of each estimator. Although limited to synthetic data, the experimental results strongly support the theoretical claims.

**Methods And Evaluation Criteria:**

The methodology is carefully constructed:
- The formalization of shared-state interference is novel and expressive, capturing dynamic systems where individuals influence and are influenced by a global state.
- The use of DML with orthogonalization and bias correction is well-motivated.
- The paper defines clear estimands (ADE and GATE), and designs plug-in and debiasing terms to achieve robustness.
- Simulation settings reflect realistic assumptions (e.g., market congestion, algorithmic competition).

Evaluation is primarily based on simulation, which is reasonable given the lack of real-world counterfactuals.

**Other Comments Or Suggestions:**

NA

**Other Strengths And Weaknesses:**

**Strengths:**
- Novel and realistic modeling of shared-state interference
- Theoretical rigor and clear assumptions
- Methodologically sound extension of DML
- Strong simulation results
- Clean writing and logical structure

**Weaknesses:**
- No real-world experiments, though the authors acknowledge this.
- Some assumptions (e.g., availability of auxiliary data) may not always hold in practice.
- The setup assumes known m in m-dependence, which could be hard to estimate in some applications.

**Questions For Authors:**

1. **Estimating m in Practice**: While the paper assumes known m for m-dependence, in real-world experiments m may be unknown. Can your method accommodate estimation of m, or how sensitive are results to misestimation?

2. **Auxiliary Data Requirement**: The use of an auxiliary dataset for nuisance estimation avoids dependence issues, but may not always be feasible. Could you discuss alternative strategies, such as block-splitting or approximate independence?

3. **Applicability to Recommender Systems**: Can your shared-state model be concretely instantiated in modern large-scale recommender systems, such as those used in e-commerce or social media? Have you explored any such datasets?

Clarifying these would enhance the paper’s applicability and help practitioners adopt your framework.

**Relation To Broader Scientific Literature:**

The paper is well-situated in the literature:
- Builds on Chernozhukov et al. (2018) for DML
- Extends recent work on interference in experiments (e.g., Johari et al., Farias et al., Munro 2024)
- Addresses limitations of social network-based interference models, offering a complementary approach via global state dynamics

The shared-state framework fills a notable gap in modeling real-world algorithmic and market-based interference.

**Theoretical Claims:**

Theoretical contributions are central to this paper:
- The paper extends DML to settings with shared-state interference by leveraging Markov chain properties.
- Theorems 3.1, 4.1, and 5.1 establish conditions for asymptotic normality of the ADE and GATE estimators.
- The variance expressions are derived under both geometric ergodicity and m-dependence.

The proofs are detailed, with attention to assumptions like Neyman orthogonality, Gateaux derivatives, and mixing conditions. The adaptation of CLT results for Markov chains is particularly well-handled.

---

> ### Author Rebuttal · Authors · 2025-03-30
>
> Thank you for your constructive feedback! We also thank you for noting the expressivity of our formality, relevance to practical settings and gap in research on causal inference in algorithmic systems and markets. We respond to each of your questions below:
> * **Estimating $m$ in Practice**: We note that for the procedure to be valid, we just need an upper bound on $m$, so that the switch period can be chosen to be greater than this upper bound. The literature on switchback experiments proposes procedures to estimating $m$ if it is not known. See, e.g., Bojinov et al 2022, Section 4.4. Informally, the procedure involves computing the estimator using different candidate values $m$ and running a series of hypothesis tests to see whether the results are the same for different values. If the results are different, then the smaller candidate for $m$ can be shown to be less than its true value, with high probability. The uncertainty introduced by unknown $m$ can be incorporated in our high-probability bound by choosing $\gamma$ to account for the Type 2 error of estimating $m$ is smaller than it actually is.
> * **Auxiliary Data requirement**: We will add more discussion of this point to the paper. It would be a worthwhile direction for future research to explore how approximately independent data (such as blocks observed far away) may be used to train nuisance predictors that are approximately independent from the data they are evaluated on.
> * **Applicability to Recommender Systems**: Application of our framework to real-world empirical contexts would be a valuable direction for future work. Given the theoretical focus of the paper and the constraints of space, we don’t do this. We strongly believe that our framework is applicable to relevant empirical settings.
>
> I. Bojinov, D. Simchi-Levi, and J. Zhao. Design and Analysis of Switchback Experiments, Apr. 2022. URL http://arxiv.org/abs/2009.00148. arXiv:2009.00148 [stat].

---

### Decision · Program_Chairs · 2025-05-01

**Decision:**

Accept (poster)

**Comment:**

The paper studies causal inference in settings where different units/individuals have shared state variables that affect them and are affected by them. It formalizes a modeling framework for this setting that determines the effect of the units/individuals on the shared state and vice versa over time, that is the units/individuals arrive sequentially and interfere through a shared Markovian state. It also provides extensions of double machine learning theorems for achieving efficient semi-parametric inference, which are then applied to estimating the average direct effect and the global average treatment effect.

The reviewers found the theoretical claims of the paper to be well-supported with clear arguments. The simulation results also support the theoretical claims and show better performance over the baselines; however, they are limited and only capture synthetic settings and not real examples or data sets. As the reviewers pointed out and the authors responded, I recommend that the authors add further discussion and remarks about the practicality and limitations of the assumptions such as the existence of auxiliary data or knowledge about ‘m’ in m-dependence.